# Mechanisms governing target search and binding dynamics of hypoxia-inducible factors

**Yu Chen**[1,2,3], **Claudia Cattoglio**[1,2,3], **Gina M Dailey**[1,3], **Qiulin Zhu**[1,3], **Robert Tjian**[1,2,3]*, **Xavier Darzacq**[1,3]*

[1]Department of Molecular and Cell Biology, University of California, Berkeley, Berkeley, United States; [2]Howard Hughes Medical Institute, University of California, Berkeley, Berkeley, United States; [3]Li Ka Shing Center for Biomedical & Health Sciences, University of California, Berkeley, Berkeley, United States

**Abstract** Transcription factors (TFs) are classically attributed a modular construction, containing well-structured sequence-specific DNA-binding domains (DBDs) paired with disordered activation domains (ADs) responsible for protein-protein interactions targeting co-factors or the core transcription initiation machinery. However, this simple division of labor model struggles to explain why TFs with identical DNA-binding sequence specificity determined in vitro exhibit distinct binding profiles in vivo. The family of hypoxia-inducible factors (HIFs) offer a stark example: aberrantly expressed in several cancer types, HIF-1α and HIF-2α subunit isoforms recognize the same DNA motif in vitro – the hypoxia response element (HRE) – but only share a subset of their target genes in vivo, while eliciting contrasting effects on cancer development and progression under certain circumstances. To probe the mechanisms mediating isoform-specific gene regulation, we used live-cell single particle tracking (SPT) to investigate HIF nuclear dynamics and how they change upon genetic perturbation or drug treatment. We found that HIF-α subunits and their dimerization partner HIF-1β exhibit distinct diffusion and binding characteristics that are exquisitely sensitive to concentration and subunit stoichiometry. Using domain-swap variants, mutations, and a HIF-2α specific inhibitor, we found that although the DBD and dimerization domains are important, another main determinant of chromatin binding and diffusion behavior is the AD-containing intrinsically disordered region (IDR). Using Cut&Run and RNA-seq as orthogonal genomic approaches, we also confirmed IDR-dependent binding and activation of a specific subset of HIF target genes. These findings reveal a previously unappreciated role of IDRs in regulating the TF search and binding process that contribute to functional target site selectivity on chromatin.

*For correspondence:
jmlim@berkeley.edu (RT);
darzacq@berkeley.edu (XD)

## Editor's evaluation

This work on dissecting the function of transcription factors using single-molecule methods is likely to appeal to the broad *eLife* readership and the gene regulation community at large. In particular, the role of transcription activation domains in transcriptional specificity is a timely contribution to the field.

## Introduction

Sequence-specific transcription factors (TFs) are key frontline regulators of gene expression. Classical LexA-Gal4 domain-swap experiments in yeast presented a simple modular structure and apparent division of labor for typical TFs (*Brent and Ptashne, 1985*). In this textbook paradigm, the DNA-binding

domain (DBD) is responsible for DNA sequence recognition and binding specificity while the activation domain (AD) is responsible for target gene transactivation that involves protein-protein interactions with co-factors, the basal transcription machinery, and other ancillary factors that are generally devoid of sequence-specific DNA recognition. In higher eukaryotes, each DBD class usually contains multiple closely related family members. For example, the bHLH class of TFs includes MyoD, Clock, and Max. They all recognize the same E-box DNA-binding sequence motif 5'-CACGTG-3', yet each differentially regulates muscle differentiation, circadian rhythm, and cell proliferation, respectively (*Kribelbauer et al., 2019*). This raises the specificity paradox: how do TFs with seemingly identical DNA sequence specificity, at least as determined in vitro, nevertheless exhibit non-overlapping binding profiles in vivo and carry out distinct and even opposing functions? In general, when confronted with this conundrum, we have assumed that one or more co-factors or perhaps still to be identified 'silent partner' TFs can somehow divert target recognition to a composite cis-regulatory site distinct from the canonical DNA-binding site. Given the high occurrence of short binding motifs for most TFs throughout the genome, even with co-operative binding to composite sites, most potential specific binding sites nevertheless remain unoccupied as determined by genome-wide TF binding studies. What feature or motif within TFs outside of the DBD and dimerization domain may be responsible for such differential site selection has remained unclear. Thus, the simple rule of modular units with well separated divisions of labor between DBD, dimerization, and transactivation may deserve a closer look. We also wondered whether quantitative single molecule dynamics measurements might reveal new aspects of TF behavior in living cells that could inform us regarding potential mechanisms influencing the target search and binding process and differential site selectivity in a native physiologically relevant context.

Here, we have chosen the hypoxia-inducible factors (HIFs) as a representative example to study the paradox of highly conserved DBDs carrying out distinct target site selection and to dissect potential novel features of TFs that mediate chromatin binding. HIFs are a family of α/β heterodimeric TFs stabilized under hypoxic conditions to promote angiogenesis, anaerobic metabolism, cell proliferation, and 'stemness' (*Semenza, 2012*). The oxygen-labile alpha subunits (mainly HIF-1α and HIF-2α) complex with their oxygen-stable beta partner (mainly HIF-1β) to form a functional dimer (*Figure 1A*). All HIF subunit isoforms belong to the bHLH-PAS (basic helix-loop-helix-PER-ARNT-SIM) family, where the N-termini are structured domains containing bHLH (DNA-binding) and PAS (dimerization) domains, while the C-termini consist of intrinsically disordered regions (IDRs) containing ADs (*Figure 1A*; *Figure 1—figure supplement 1A*). HIF-1α/1β and HIF-2α/1β dimers share a conserved structural fold (*Wu et al., 2015*), recognize the same hypoxia response element (HRE) 5'-TACGTG-3' binding motif (*Schödel et al., 2011*; *Wenger et al., 2005*), but share only a partial overlap of target genes in vivo (*Smythies et al., 2019*). With their own unique target gene sets, HIF-1α and -2α can exert divergent and even contrasting functions (*Keith et al., 2012*). For example, while both HIF-1α and HIF-2α regulate angiogenesis, HIF-1α specifically regulates glycolysis, apoptosis, and promotes NO production, whereas HIF-2α binds to the *POU5F1* locus to maintain Oct4-regulated stem cell identity and pluripotency, promotes cell cycle progression, and inhibits NO production (*Keith et al., 2012*; *Smythies et al., 2019*). Therefore, our current simple textbook model of exchangeable modular TF functional units does not satisfactorily explain such isoform-specific target gene regulation.

The HIF family differential specificity paradox is even more daunting to comprehend at the level of disease inducing mechanisms. HIFs are aberrantly upregulated and recognized as oncogenic drivers in multiple cancers. However, in addition to their shared roles in cancer onset and progression, HIF-1α and -2α also show many independent, sometimes even opposing roles in specific contexts (*Keith et al., 2012*). For example, in clear cell renal cell carcinoma (ccRCC), HIF-2α is the critical tumorigenic driver whereas HIF-1α, in contrast with its usual tumorigenic role, is mostly tumor-suppressive (*Raval et al., 2005*; *Schödel et al., 2016*). The regulatory mechanism behind such highly divergent outcomes is still largely unknown. Given such complexity, without a deeper understanding of isoform-specific transcriptional regulation, it is hard to predict the functional outcomes mediated by individual HIF isoforms in various cancer types or stages, which could be a complicating factor in developing more effective HIF-targeting cancer therapeutics.

In this study, we aim to understand the molecular mechanisms mediating isoform-specific target gene regulation at its most fundamental level – could we detect differential molecular dynamics of distinct TF isoforms during the target search and chromatin binding process in live cells? Which regions or domains of TFs might be responsible for such isoform-specific properties? Could we begin to

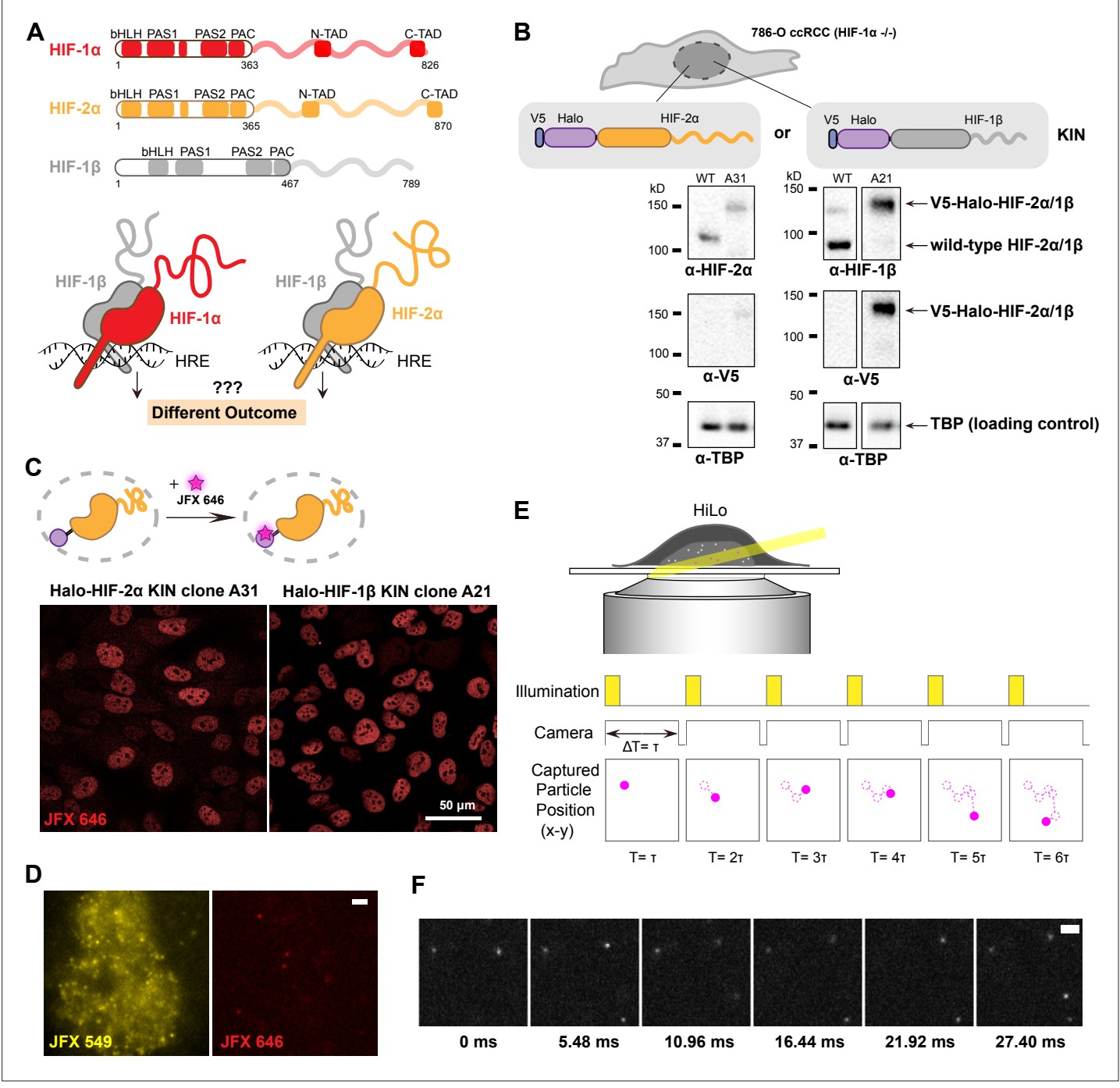

**Figure 1.** Endogenous tagging of hypoxia-inducible factors (HIFs) in 786-O clear cell renal cell carcinoma (ccRCC) cells for fast single particle tracking (fSPT). (**A**) Schematic showing the similar domain organizations of HIFs (top) and the HRE-bound HIF α/β dimers (bottom). Disordered regions are represented as wavy lines. (**B**) Generation of Halo KIN clones in the HIF-1α negative 786-O ccRCC line. Top: Halo-tagging scheme of HIF-2α (left) and HIF-1β (right). Bottom: Western blot of wild-type (WT) 786-O cells and homozygously tagged knock-in clones (A31 and A21). Panels derive from the same blots shown in full in *Figure 1—figure supplement 2*. See *Figure 1—source data 1* for images of the raw blots. (**C**) Halo-tagged HIF-2α and HIF-1β show predominant nuclear localization. Top: Schematic of labeling Halo-tagged proteins in live cells with cell-permeable Halo-binding JFX646 dye. Bottom: Representative images of Halo-HIF-2α (left) and Halo-HIF-1β (right) clones labeled with 500 nM JFX646 (**D**) representative images showing the same cell labeled with a high concentration of JFX549 dye for localizing the nucleus in one channel (left) and labeled sparsely with JFX646 dye for tracking individual molecules in another channel (right). Scale bar = 2 μm. (**E**) Graphical illustration of fSPT capturing trajectories of moving particles. Top: Highly inclined and laminated optical sheet illumination (HiLo). Bottom: Illumination and camera sequence with corresponding particle position

*Figure 1 continued on next page*

*Figure 1 continued*

at each frame (solid magenta dots). Particle's past positions (dashed magenta circles) are connected with dotted magenta lines to show the particle's trajectory. (**F**) Actual data showing detection of Halo-HIF-2α protein molecules at 5.48 ms frame rate. Scale bar = 2 µm.

The online version of this article includes the following source data and figure supplement(s) for figure 1:

**Source data 1.** Source data for both *Figure 1* and *Figure 1—figure supplement 2*, including six raw images (tif files) for western blots and one figure (pdf) explaining each of the raw images.

**Figure supplement 1.** Domain analysis of hypoxia-inducible factors (HIFs).

**Figure supplement 2.** Verification of endogenous tagging of hypoxia-inducible factors (HIFs) in 786-O clear cell renal cell carcinoma (ccRCC) cells.

**Figure supplement 3.** Hypoxia-inducible factor (HIF) genome-wide binding and RNA expression profiles after gene-editing and genetic modification.

discern possible mechanisms that guide TFs with highly conserved DBDs to their distinct and specific targets beyond cognate DNA sequence recognition? Here, we use HIFs as an illustrative example, combining endogenous tagging and super-resolution single particle tracking (SPT) (*Liu et al., 2015*) to study the dynamic behavior of these key gene regulators in live cells under physiological conditions. We also dissect the contribution of different domains of HIF-α isoforms by a series of mutation and domain-swap experiments to directly test the concept of modular functional domains. Deploying a combination of genetic and small-molecule perturbations, we found that, although HIF DBD and dimerization are important for DNA target acquisition, differences between HIF isoforms in terms of amount of protein bound and diffusion characteristics are mainly driven by regions outside the DBD and dimerization domains. Finally, using genomic approaches we found that, in concordance with our imaging results, binding strength and gene activation are IDR-dependent for a subset of HIF target sites. Our results reveal a previously unappreciated role of unstructured domains in the target search and binding properties of TFs to functional chromatin sites in a live cancer cell context.

## Results

### Establishing a human cancer cell system for live-cell single molecule imaging of HIF

To investigate HIF dynamics, we first focused on one of the cognate dimers: HIF-2α/1β. We used the common ccRCC line 786-O (*Brodaczewska et al., 2016*), derived from a VHL-deficient, H2 type primary ccRCC, wherein HIF-2α is stabilized due to an inactivating mutation in VHL (the E3 ubiquitin ligase that targets all HIF-α isoforms for proteasomal degradation) (*Gnarra et al., 1994*). The 786-O line also conveniently lacks any functional HIF-1α due to a truncating mutation of HIF-1α (*Shen et al., 2011*; *Swiatek et al., 2020*), which allows us to study one α isoform independently from the other. Using CRISPR/Cas9-mediated genome editing, we successfully generated several clonal lines with homozygous knock-in (KIN) of the HaloTag (*Los et al., 2008*) at the N-terminus of either HIF-2α or its binding partner, HIF-1β (*Figure 1B*; *Figure 1—figure supplement 2*). Western blotting confirmed that the tagged proteins are expressed at levels similar to wild-type (WT) in unedited cells (e.g. HIF-2α clone A31 and HIF-1β clone A21) (*Figure 1B*). Confocal imaging after covalently labeling cells with a fluorescent Halo-binding ligand (JFX646) (*Grimm et al., 2021*) shows the expected nuclear localization for both Halo-HIF-2α and Halo-HIF-1β proteins (*Figure 1C*). In addition, we confirmed by ChIP-seq that both tagged proteins maintain a similar genome-wide binding profile as the WT protein in unedited cells (*Figure 1—figure supplement 3A*), and RNA-seq confirmed that gene expression profiles in both edited cell lines are not significantly altered from WT unedited cells (*Figure 1—figure supplement 3B–C*, with cells overexpressing WT and mutant HIF-α as controls). We have thus established a human cancer cell system suitable for live-cell imaging of HIF-2α and HIF-1β at endogenous expression levels.

To evaluate how HIF-2α and -1β explore the nucleus and bind DNA, we used the fast modality of super-resolution live-cell SPT (fast SPT, or fSPT) that is capable of tracking rapidly diffusing molecules. Cells with either HIF-2α or HIF-1β Halo KIN were doubly labeled with the live-cell permeable Halo-binding JFX dyes (*Figure 1D*) and were imaged under highly inclined and laminated optical sheet illumination (HiLo) (*Tokunaga et al., 2008*) at high frame rates (~182 Hz) to capture the movement of single molecules in their native nuclear environment (*Figure 1E*). Stroboscopic illumination at high excitation power is used to minimize motion blur (*Elf et al., 2007*; *Hansen et al., 2017*; *Hansen*

*et al., 2018*), while sparse labeling ensures that only a limited number of molecules are detected at any given time in the nucleus to minimize misconnections when computing the path of individual molecules (trajectories) (*Figure 1E–F*). We can then estimate relevant kinetic parameters from these trajectories, extracting quantitative information such as diffusion coefficients and bound fraction.

## fSPT detects various HIF molecular states in their native nuclear environment

To quantitatively analyze the acquired fSPT data, we used a non-parametric Bayesian 'state array' (SA) approach recently developed in our lab, which estimates the occupations of a range of dynamic states with diffusion rates from 0.01 to 100 μm$^2$/s while accounting for known experimental biases due to localization error and fluorophore defocalization (*Heckert et al., 2022*). Briefly, SA analyzes regular Brownian motion with normally distributed localization error (RBME) with a two-dimensional 'grid of states' that spans a range of diffusion coefficients (first dimension) and localization error (second dimension) magnitudes, and evaluates the occupancy (proportion) of each state in this array using the observed fSPT trajectory data and a variational Bayesian algorithm. After inference, the proportion of molecules as a function of diffusion coefficient is calculated by marginalizing localization error, creating a 'diffusion spectrum' that reports the underlying molecular subpopulations according to their proportions (y-axis) and their diffusive properties (diffusion coefficients, x-axis) (*Figure 2—figure supplement 1*). While our previous approach, which already addressed biases due to localization error and defocalization, required the assumption of a fixed and limited (3 or less) number of states for the tracked protein (*Hansen et al., 2018*), our current SA approach does not require a priori knowledge or any assumption regarding the number of subpopulations or 'states' of the tracked molecules. We chose to use this new method, because as dimeric TFs, HIF subunits could conceivably exist in many states (e.g. bound, moving as a monomer, moving as a dimer, and moving in a bigger complex containing co-regulators) (*Figure 2A–B*) and the SA approach better suits our needs for model-free analysis without assuming any given number of states.

We first analyzed individual cells to check cell-to-cell variation. Due to the small number of trajectories per cell, we used a less precise version of the SA calculation, the 'naïve occupation estimate' by simply applying the RBME likelihood estimation to individual cells without refining the posterior over diffusion coefficient and localization error (*Figure 2—figure supplement 1A–B*, top) (*Heckert et al., 2022*). We observed rather heterogeneous results for both HIF-2α and HIF-1β, with varied diffusion coefficient estimates from cell to cell (*Figure 2A–B*; *Figure 2—figure supplement 2A*, clones A31 and A21). While the heterogeneity we observed was likely due to the limited number of trajectories collected from each cell (i.e. small sample bias) rather than to a difference in behavior of HIF proteins in each cell, our findings do indicate that a range of moving states likely exists for both HIF proteins.

We then pooled trajectories from many cells (n~60 from three biological replicates for each experimental condition) to estimate the distribution of diffusion coefficients for the population. SA generates a distribution of diffusion coefficient estimates that reports the fraction of stably bound molecules while simultaneously displaying the full behavioral spectrum of the diffusing molecules (*Figure 2C*; *Figure 2—figure supplement 1A–B*, bottom). We define the fraction with a diffusion coefficient <0.1 μm$^2$/s as immobile and presumably chromatin-bound (see *Figure 2—figure supplement 3* and Appendix 1 for discussion of source of variations between replicates). Strikingly, we observed a very different behavior for Halo-HIF-2α compared to Halo-HIF-1β. Whereas a large fraction (about 40%) of Halo-HIF-2α is bound, the majority (above 70%) of Halo-HIF-1β appears freely diffusing (*Figure 2C*; *Figure 2—figure supplement 2B*, clones A31 and A21). Also, the overall diffusion coefficient for the Halo-HIF-1β mobile population is much larger than that of Halo-HIF-2α. We repeated measurements in different KIN clones and confirmed the reproducibility of these results for both Halo-HIF-2α (*Figure 2—figure supplement 2A–B*, clone B50) and Halo-HIF-1β (*Figure 2—figure supplement 2A–B*, clone B89). Note that although we quantitatively compared the bound fraction, due to the complexity of the composition of the moving population (i.e. multiple states might exist but are merged into a single peak), quantitatively comparing mode (i.e. peak) or mean diffusion coefficient may give slightly different results. Therefore, for the rest of the paper we only present the peak diffusion coefficient in the figures but listed both peak diffusion coefficient and mean diffusion coefficient for the moving population in *Supplementary file 1*.

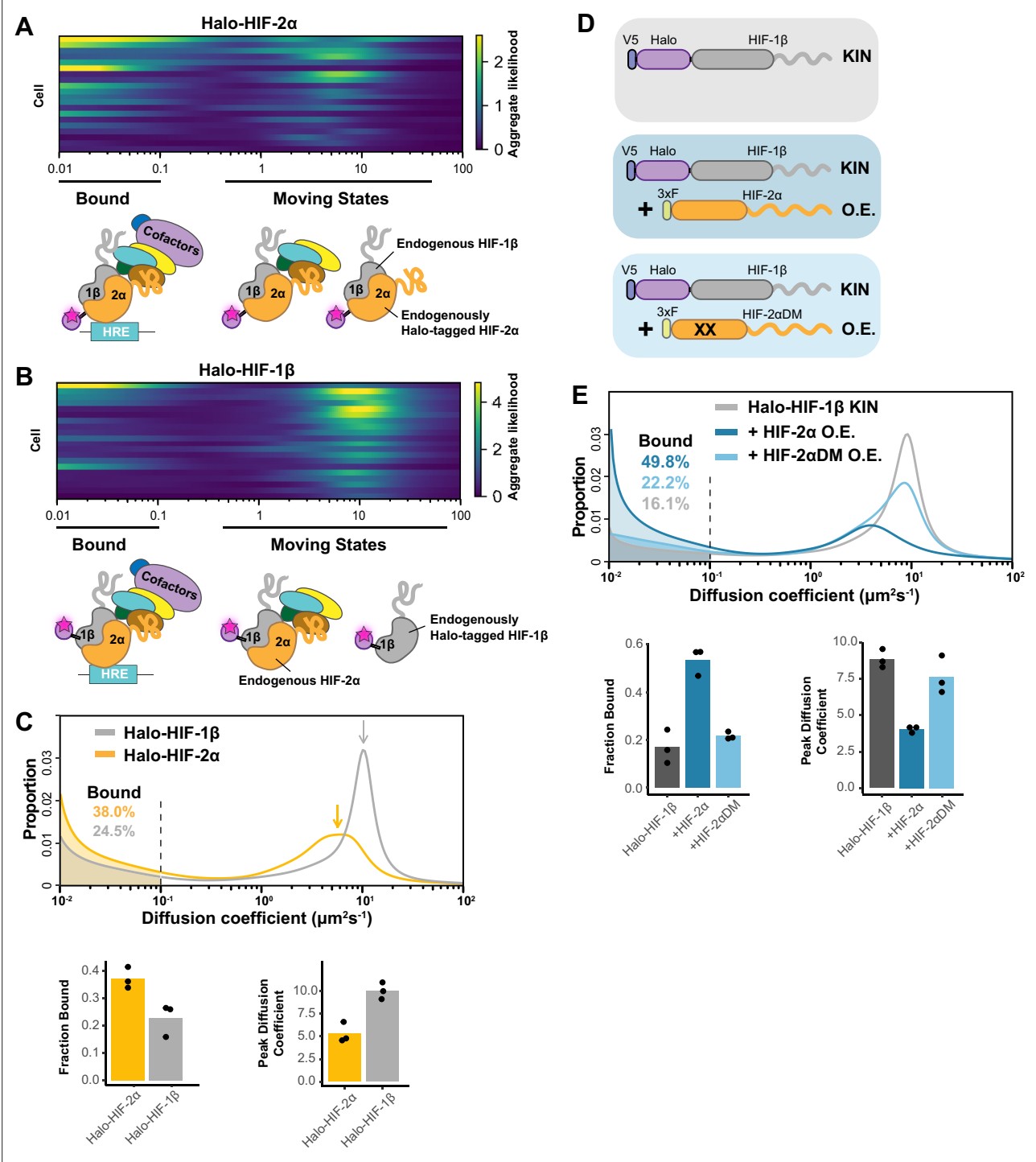

**Figure 2.** Fast single particle tracking (fSPT) sensitively detects molecules in a range of states. (A–B) Likelihood of diffusion coefficients based on a model of regular Brownian motion with normally distributed localization error (RBME) (*Heckert et al., 2022*), for (**A**) Halo-HIF-2α, clone A31 and (**B**) Halo-HIF-1β, clone A21, with drawing illustrating bound and different hypothetical moving states: complexes, dimer and monomer. Each row represents data collected from one cell. 0.1 μm²/s is used as the cut-off for bound versus free. (**C**) Top: Proportion of molecules as a function of their diffusion coefficients (posterior mean occupations for a state array; *Heckert et al., 2022*) evaluated on trajectories across all cells measured for each knock-in(KIN) line (Halo-HIF-2α, clone A31 and Halo-HIF-1β, clone A21). Compared to HIF-2α, HIF-1β has less bound fraction (gray versus yellow shaded areas) and faster diffusion coefficient (gray versus yellow arrows). Bottom: Summary of the bound fraction (left) and peak diffusion coefficient (right) for the two clones. Each bar represents the averaged value from three independent measurements on different days (black dots). *Supplementary file 1* reports cell and trajectory numbers for this and all the following figures. (**D–E**) Overexpressing HIF-2α, but not a dimerization mutant form, in the Halo-HIF-1β

*Figure 2 continued on next page*

*Figure 2 continued*

knock-in (KIN) line increases HIF-1β binding and decreases its diffusion coefficient. (**D**) schematic illustrating the parental Halo-HIF-1β KIN cells (gray background) and cells stably overexpressing (O.E.) either the wild-type (darker blue background) or a dimerization mutant (DM, black crosses, lighter blue background) form of HIF-2α. (**E**) Top: Proportion of molecules as a function of diffusion coefficient measured for HIF-1β in Halo-HIF-1β KIN cells (gray) and in Halo-HIF-1β KIN cells overexpressing HIF-2α (WT, dark blue background, or dimerization mutant (HIF-2αDM), light blue). Shaded areas indicate bound fraction. Bottom: Bar plot of the average value (bar height) of the bound fraction (left) and peak diffusion coefficient (right) calculated from three independent measurements (black dots) for each condition.

The online version of this article includes the following figure supplement(s) for figure 2:

**Figure supplement 1.** State array (SA) analysis on fast single particle tracking (fSPT) data.

**Figure supplement 2.** HIF-1β behavior is distinct from HIF-2α and changes as 1β-to-2α stoichiometry changes.

**Figure supplement 3.** Analysis of source of variation by bootstrapping.

The differences between HIF-2α and -1β seem counterintuitive at first, because one would expect HIF-2α and HIF-1β to behave similarly since they should exist as a heterodimer. However, since the endogenous HIF-1β is expressed at a much higher level than HIF-2α (*Figure 1—figure supplement 2C*), the majority of HIF-1β is likely free to diffuse without HIF-2α. Of note, the distribution plot only reflects the fraction of molecules as a function of their diffusion coefficient, but does not report on the absolute number of molecules. Therefore, a smaller bound fraction for Halo-HIF-1β does not mean fewer numbers of bound molecules than Halo-HIF-2α, since many more Halo-HIF-1β molecules are present in the nucleus. Given this scenario, we hypothesized that HIF-1β molecular dynamics and percent binding should be modulated by changing the 2α/1β stoichiometry.

## HIF-1β binding and diffusing dynamics can be modulated by HIF-α:β stoichiometry, and are dependent on dimerization

To test the hypothesis that HIF-1β dynamics depends on 2α/1β ratio, we first tried to modulate its behavior by stably overexpressing HIF-2α in the endogenously HIF-1β Halo-tagged KIN line (*Figure 2D*). We found that the mobile population of Halo-HIF-1β diffuses more slowly when HIF-2α is overexpressed, most likely due to its dimerization with the extra HIF-2α to form dimers capable of DNA/chromatin binding. As expected, we also observed a significant increase in the Halo-HIF-1β bound fraction (up to 50%), (*Figure 2E*; *Figure 2—figure supplement 2C*, top and middle). To confirm that the changes in HIF-1β dynamics caused by increasing levels of HIF-2α are dependent on heterodimerization, we stably overexpressed the HIF-2α R171A/V192D double mutant (HIF-2α DM) that was previously reported to lose its dimerization capability with HIF-1β (*Wu et al., 2015*). As expected, overexpression of HIF-2α DM did not increase the bound fraction or decrease the overall diffusion coefficient of Halo-HIF-1β to the same extent seen with WT HIF-2α overexpression (*Figure 2E*; *Figure 2—figure supplement 2C*, bottom), suggesting that the changes we observe are dimerization-dependent.

We further validated our results by stably overexpressing different forms of HIF-α in the HIF-1β Halo-tagged KIN line and treating cells with an HIF-2α-specific small-molecule inhibitor, Belzutifan (PT-2977). Belzutifan inhibits HIF-2α/1β, but not HIF-1α/1β, dimerization by specifically binding to the dimerization domain of HIF-2α (*Figure 3—figure supplement 1A*), and thus has been used as an HIF-2α inhibitor for ccRCC treatment (*Wallace et al., 2016*; *Xu et al., 2019*). We first confirmed that Belzutifan inhibits HIF-2α transcription function in a dose-dependent manner (*Figure 3—figure supplement 1B*). Importantly, Belzutifan also reduces the HIF-2α bound fraction in the HIF-2α Halo-tagged KIN line in a similar dosage-dependent manner, again revealing the potential of fSPT to measure TF dynamics and associated functional changes (*Figure 3—figure supplement 1C–E*). We choose to use 0.2 μM Belzutifan for all subsequent experiments to maximize its effect.

Next, we carried out a series of experiments designed to probe the consequences of swapping different functional domains of HIF-1α and HIF-2α to determine which parts of these closely related TFs might be involved in selective activities when paired with HIF-1β. Using the HIF-1β Halo-tagged KIN line as the parental line, we stably overexpressed WT or chimeric HIF-α, where we swapped the structured and disordered domains between HIF-1α and HIF-2α (*Figure 3A*). All these different HIF-α variants are expressed from a relatively strong EF-1α promoter and are N-terminally 3xFLAG-tagged. A construct that expresses 3xFLAG only is used as control. We then treated these cells with either

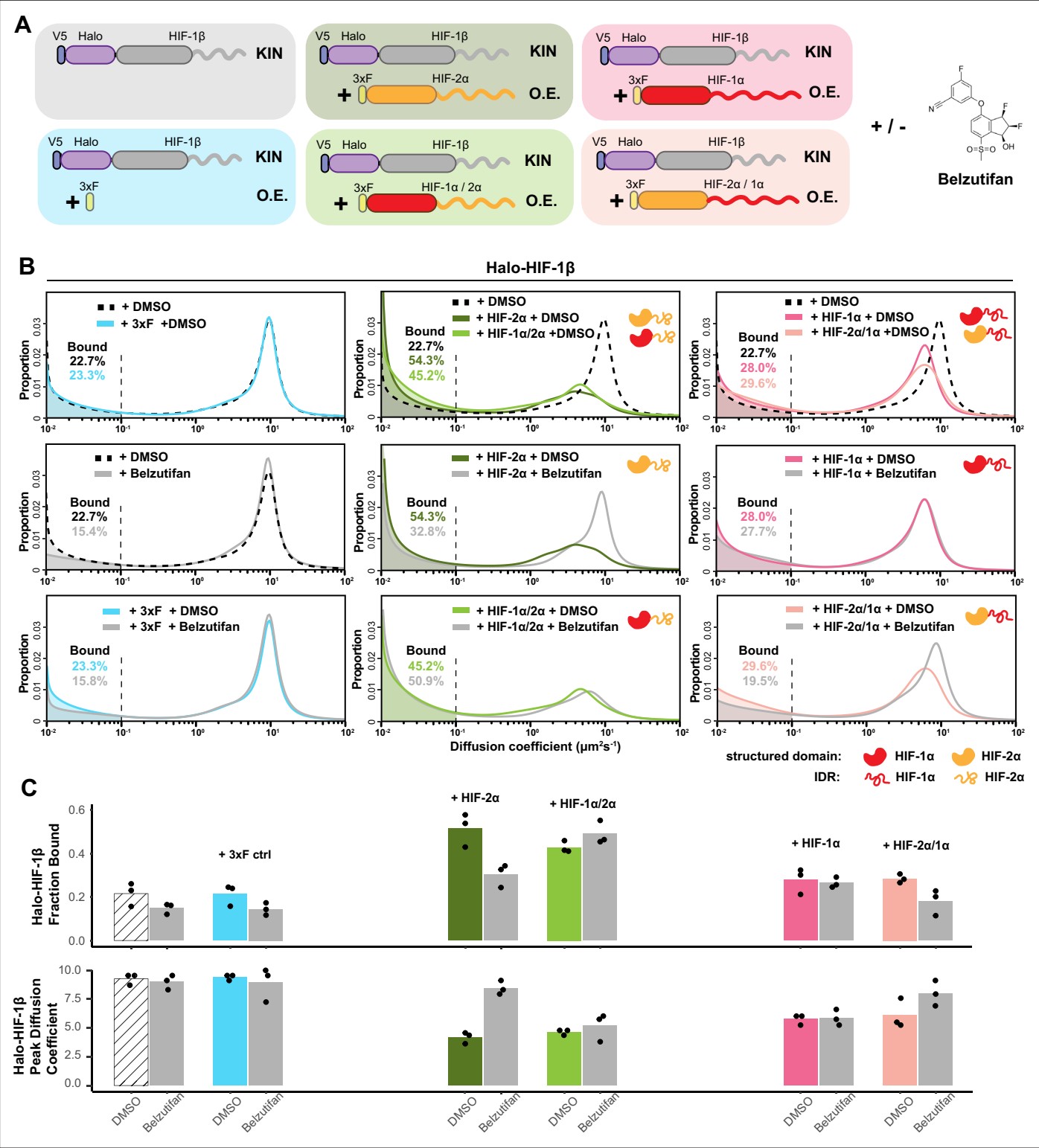

**Figure 3.** HIF-α increases HIF-1β binding and decreases HIF-1β diffusion coefficient through dimerization, in an intrinsically disordered region (IDR)-dependent manner. (**A**) Schematic of evaluating Halo-HIF-1β behavior with genetic and small-molecule perturbation. Parental Halo-HIF-1β knock-in (KIN) cells (gray background) and cells stably overexpressing (O.E.) either a certain form of HIF-α (wild-type [WT] or domain swap, HIF-1α, red, HIF-2α, orange. Disordered regions are represented as wavy lines.) (various colored background), or a 3xFLAG tag only control (blue background) are used, with and without 0.2 µM Belzutifan (HIF-2α/1β dimerization inhibitor) treatment. (**B**) Proportion of Halo-HIF-1β molecules as a function of diffusion coefficient measured in various conditions outlined in (**A**) Top row: DMSO only, showing overexpressing α subunit can change HIF-1β behavior. Cells

*Figure 3 continued on next page*

*Figure 3 continued*

overexpressing the α subunit variants containing HIF-2α disordered region (orange curly line) have a stronger effect (middle, HIF-2α and HIF-1α/2α,) compared to those containing HIF-1α disordered region (right, HIF-1α and HIF-2α/1α). Middle and bottom rows: Proportions of HIF-1β as a function of diffusion coefficient, measured in each of the six cell lines with Belzutifan treatment are compared to the DMSO control. Changes caused by overexpressing an α subunit can be specifically reverted by Belzutifan treatment for cell lines expressing an α subunit variant that contains the HIF-2α structured domain (orange globule). (**C**) Summary of the average bound fractions (top) and peak diffusion coefficient (bottom) for all 12 conditions, with black dots indicating values from each of the three individual measurements.

The online version of this article includes the following figure supplement(s) for figure 3:

**Figure supplement 1.** Dosage-dependent inhibition of HIF-2α binding and activity by Belzutifan.

**Figure supplement 2.** Additional data for *Figure 3* showing HIF-α without the intrinsically disordered region (IDR) is not able to increase HIF-1β binding or decrease its diffusion coefficient.

Belzutifan or DMSO control and measured Halo-HIF-1β dynamics (*Figure 3A*). While 3xFLAG tag had no effect, overexpressing HIF-α, regardless of which variant form, is able to both increase the bound fraction and reduce the overall diffusion coefficient of HIF-1β (*Figure 3B*, top, and *Figure 3C*, DMSO group). For cells overexpressing the α variants that contain the HIF-2α structured domain, this effect on HIF-1β can be at least partially reverted after Belzutifan treatment (*Figure 3B–C*, +HIF-2α and +HIF-2α/1α). In contrast, for cells overexpressing the α variants that contain the HIF-1α structured domain, this effect is resistant to Belzutifan, consistent with the subunit isoform specificity of the drug for HIF-2α (*Figure 3B–C*, +HIF-1α and +HIF-1α/2α). In untransfected and 3xFLAG only overexpressing control cells, treatment with Belzutifan only weakly reduces the HIF-1β bound fraction, again suggesting that the majority of HIF-1β is not engaged with 2α (*Figure 3B–C*, parental cell and +3xF). Overall, these results demonstrate that HIF-1β dynamics change after engagement with its α partner and can be selectively inhibited with a specific dimerization inhibitor. The observed differences also confirm that fSPT is a powerful platform to monitor molecular dynamic changes of TFs in living cells thus, allowing us to gain new mechanistic insights while we introduce various perturbations, such as subunit concentration or stoichiometry and specific mutations.

## Regions outside the DBD/dimerization domain determine HIF molecular dynamics

Interestingly, comparing the effects of the four different α variants, we found that regardless of their structured domain, those with the same C terminal IDRs behave similarly (*Figure 3B–C*, middle and right). Specifically, the variants containing the HIF-2α IDR have a stronger effect on increasing HIF-1β binding than the variants containing the HIF-1α IDR. Thus, surprisingly, it appears that the bound fraction of HIF-1β is not determined by the HIF-α DBD, but rather by HIF-α IDR, which we found rather counterintuitive. To confirm the importance of HIF-α IDRs in HIF binding, we overexpressed a truncated version of either HIF-1α or -2α that contains only the N-terminal structured region (HIF-1α NT or HIF-2α NT), which still maintains both the DBD and dimerization capability for interacting with HIF-1β (*Wu et al., 2015*). Indeed, both these truncated forms lacking the IDR/AD of HIF-α minimally affect the HIF-1β bound fraction (*Figure 3—figure supplement 2*). Surprisingly, these truncated HIF-α variants also only marginally influenced the overall HIF-1β diffusion coefficient. These results indicate that dimerization alone neither increases HIF-1β binding nor reduces the overall diffusion coefficient of its moving population. Instead, the extended HIF-α AD-containing IDR appears necessary to influence and direct HIF-1β behavior.

To further test our hypothesis that HIF chromatin binding and the dynamics of the diffusion population are dominated by the α subunit IDR, we switched to image the α subunit itself. We made different forms of Halo-tagged HIF-α (WT and domain-swapped), stably but weakly expressed them in WT 786-O cells with an L30 promoter (*Figure 4A*). To minimize differences in expression levels among different HIF-α forms, we selected cells with a roughly similar fluorescence intensity in the 'cell-picking channel' (*Figure 1D*, left), where Halo-tagged proteins were fully labeled with JFX549 to reflect their amount (except for the few molecules labeled with JFX646 for fSPT tracking) (*Figure 1D*, right). Since we achieved sparse labeling using low JFX646 dye concentrations, we did not need a pre-bleaching step before collecting SPT movies. We could thus more confidently correlate the average number of particles detected in the initial 10 frames (~50 ms) in the 'fSPT tracking channel' (*Figure 1D*, right) with expression levels (*Figure 4—figure supplement 1*; see also Appendix 2 for full

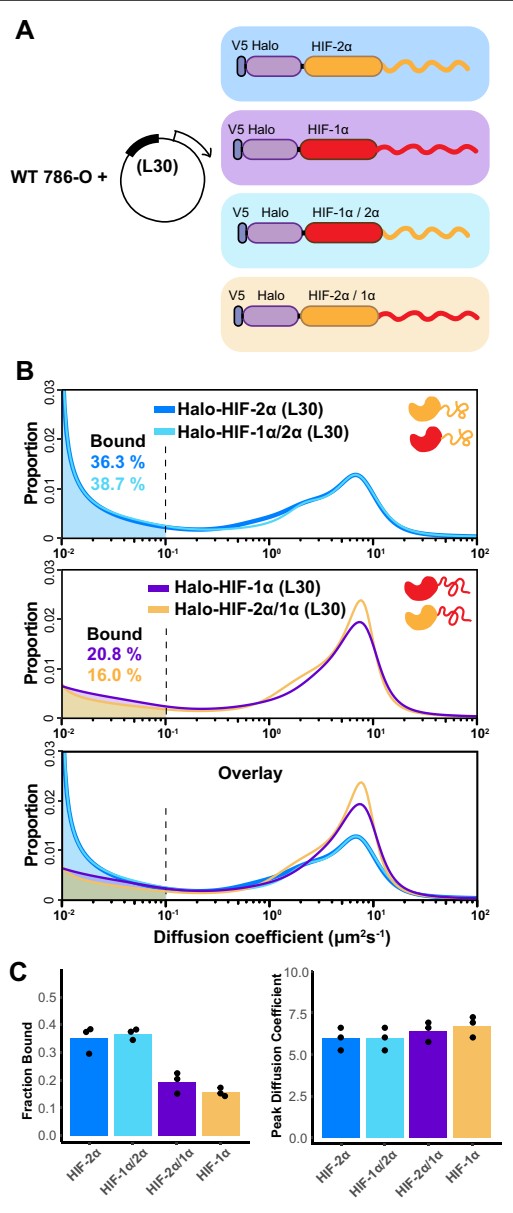

**Figure 4.** The intrinsically disordered region (IDR) governs HIF-α molecular dynamics and binding characteristics. (**A**) Schematic representation of different hypoxia-inducible factor (HIF) (wild-type [WT] and domain-swapped) being weakly and stably overexpressed with an L30 promoter and tracked in WT 786-O cells. (**B**) Proportion of molecules as a function of diffusion coefficient for every tracked protein in (**A**). Top: Overlapping distribution curves shows almost identical behavior between Halo-HIF-2α (dark blue curve) and Halo-HIF-1α/2α (light blue curve). Middle: Similar behavior between Halo-HIF-1α (purple curve) and Halo-HIF-2α/1α (yellow curve). Bottom: Overlay of all four curves shows very different behavior between proteins containing 1α versus 2α IDR. (**C**) Bar plot comparing the average bound fraction (left) and peak diffusion coefficient (right) for cells in (**B**), with

*Figure 4 continued*

black dots indicating values from three independent measurements.

The online version of this article includes the following source data and figure supplement(s) for figure 4:

**Figure supplement 1.** Quantification of initial localization density showing we imaged cells with similar expression levels of Halo-tagged proteins.

**Figure supplement 1—source data 1.** Source data for *Figure 4—figure supplement 1*, including two raw images (tif files) for western blots and one figure (pdf) explaining each of the raw images.

**Figure supplement 2.** L30 weak expression system is able to recapitulate the endogenous protein behavior.

discussion). Indeed, the initial average localization density only varied within a factor of 2 across different cell lines in each experimental group. We confirmed that, when controlling for similar expression levels, binding and diffusion characteristics of L30-expressed Halo-HIF-2α are very similar to the endogenous Halo-HIF-2α in the KIN line (*Figure 4—figure supplement 2*), demonstrating that weak overexpression can largely recapitulate endogenous protein behavior. Therefore, this system provides a convenient tool to investigate the contribution of each domain of HIF-α in the target search and binding process.

Much like our results with endogenous HIF-1β, we observed similar behaviors of HIF-α proteins if they contain the same IDR (*Figure 4B*, top and middle), while displaying distinct behaviors when endowed with different IDR isoforms (*Figure 4B*, bottom). Regardless of which DBD they have, the variants containing the HIF-2α IDR (WT HIF-2α and HIF-1α/2α) show a higher bound fraction, compared to the ones containing HIF-1α IDR (WT HIF-1α and HIF-2α/1α) (*Figure 4C*). These results suggest that indeed the disordered region on HIF-α determines how HIFs bind and diffuse in the nucleus, and that the HIF-2α AD-containing IDR mediates more or stronger binding to chromatin and/or some other relatively immobile components in 786-O cells.

## HIF-α disordered region is necessary but not sufficient for optimal binding

The fact that the extent of binding (presumably to chromatin) of HIF proteins depends mainly on the long C-terminal IDR rather than on their DBD was unexpected. Therefore, we next examined the contribution of the HIF DBD to the bound fraction. We introduced point mutations in the DBD

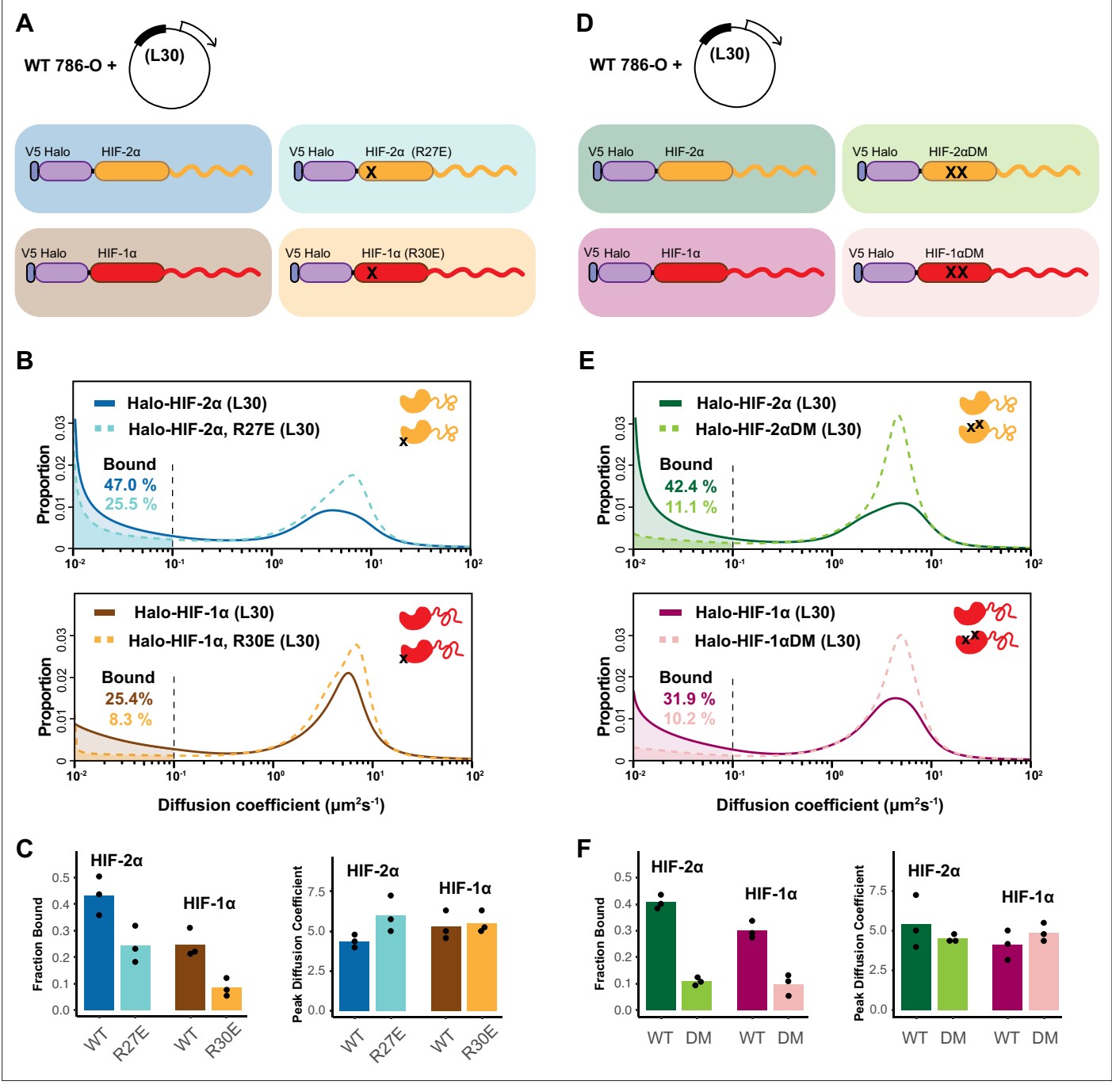

**Figure 5.** HIF-α intrinsically disordered region (IDR) alone is not sufficient for binding. (**A–C**) Mutation in DNA-binding domain (DBD) reduces the bound fraction for both HIF-α isoforms. (**A**) Schematic representation of weakly overexpressing and tracking wild-type and DBD mutant (R27E or R30E, black cross) forms of HIF-1α or -2α, using the same L30 expression system as in *Figure 4*. (**B**) Proportion of molecules as a function of diffusion coefficient for tracked protein listed in (**A**). (**C**) Bar plot summarizes the average value (bar height) of the bound fraction (left) and peak diffusion coefficient (right) of three independent measurements (black dots). (**D–F**) Mutations in the dimerization domain reduce the bound fraction for both HIF-α isoforms but do not change their diffusion coefficient. (**D**) Schematic representation of weakly overexpressing and tracking wild-type and dimerization mutant (DM, two black crosses) forms of Halo-HIF-1α or -2α, using the L30 expression system. (**E**) Proportion of molecules as a function of diffusion coefficient for tracked protein in (**D**). (**F**) Summary of the average value (bar height) of the bound fraction (left) and peak diffusion coefficient (right) for all four proteins with black dots indicating values from three independent measurements.

(HIF-2α R27E and HIF-1α R30E) that were previously reported to impair DNA binding (*Michel et al., 2002*; *Wu et al., 2015*), and expressed them in the WT 786-O cells with the same L30 promoter system (*Figure 5A*). Not surprisingly, DBD mutants show a reduction in the bound fraction and a concomitant increase in the diffusing fraction compared to their WT counterpart (*Figure 5B–C*). In agreement with the expectation that the DBD mutations should not perturb protein-protein interactions, we do not observe a significant change in the overall diffusion coefficient of the moving population. These results demonstrate that, although the AD-containing IDR is the major modulator in determining the differences in bound fraction among different HIF dimers, the DBD is also important for binding, further suggesting that the observed bound fraction likely represents chromatin/DNA binding.

We next examined whether dimerization with HIF-1β is required for HIF-α chromatin binding. Taking advantage of the same L30 weak expression system, we exogenously expressed the Halo-HIF-2α dimerization mutant (R171A/V192D), or the analogous Halo-HIF-1α dimerization mutant (R170A/V191D) in the WT 786-O cells (*Figure 5D*). We found that compared to the WT Halo-HIF-2α or -1α, these mutants exhibit a significantly decreased bound fraction (*Figure 5E–F*), demonstrating that HIF-α without -1β can no longer effectively bind to DNA/chromatin. Taken together, our results indicate that the HIF-α disordered region alone is not sufficient to maintain binding, but instead, the IDR and both the DBD and dimerization domains are also needed.

## Intrinsic properties of HIF-α IDR determine the overall rate of diffusive HIF

Interestingly, with the Halo-HIF-α dimerization mutants, we observed no obvious change in their overall diffusion coefficient in the moving population (*Figure 5E–F*; *Supplementary file 1*), indicating that losing their HIF-1β partner does not affect the overall HIF-α diffusion rate. This result suggests that it is some intrinsic property of HIF-α molecules, rather than the molecular weight of dimers versus monomers, that determines its diffusion rate and behavior. Our results suggest that while the moving population of HIF-1β alone diffuses relatively fast, the moving population of both HIF-α and HIF-α/β dimers diffuses relatively slowly. We postulate this is potentially due to the HIF-α IDR engaging in protein-protein interactions with various co-factors both when associated with HIF-1β or when alone (Figure 8A). Indeed, this is consistent with our previous observation that the HIF-α NT/HIF-1β dimer diffuses at a relatively fast rate, similar to HIF-1β alone which apparently does not share this HIF-α IDR-mediated capacity (*Figure 3—figure supplement 2*).

## HIF-α IDR contributes to binding preferences and regulatory specificity of target genes

To see how differences in HIF molecular dynamics relate to actual genome-wide binding, we performed Cut&Run on 786-O cells stably expressing different forms of Halo-tagged HIF-α driven by the L30 promoter (same cells we performed fSPT on in *Figure 4*). All these exogenously expressed proteins contain a V5 tag at the N-terminus of the Halo-tag, which we used as the epitope for Cut&Run antibody recognition. The Halo-HIF-2α KIN clone A31 served both as a positive control and as a reference for endogenous HIF-2α binding, as it also contains a V5 tag N-terminal to the Halo-tag. WT cells without genetic modification controlled for V5 antibody non-specific binding. Interestingly, the overall genome-wide binding profiles are very similar for all HIF-α variants we examined (*Figure 6—figure supplement 1*), where most of the endogenous HIF-2α binding sites are also bound by all exogenously expressed HIF-α (-1α, -1α/2α, -2α, and -2α/1α). Even genes that are known to be regulated by one or the other isoform (e.g. the HIF-1α responsive gene *PGK1* and the HIF-2α responsive gene *TGFA*; *Keith et al., 2012*) are bound by all α forms at their promoters/enhancers.

However, we did find significant differential binding preferences dictated by the IDR at a subset of HIF target sites (*Figure 6A–B*; *Figure 6—figure supplement 2*). For example, the enhancer of HIF-2α responsive genes such as *HSD3B7* and *PLXNA2* show increased HIF-2α binding when cells are expressing extra HIF-2α, but no change in HIF-1α binding when HIF-1α is overexpressed. Interestingly, such an increase in binding is also seen when overexpressing HIF-1α/2α, but not the HIF-2α/1α chimera and, importantly, results in selective gene activation (*Figure 6B*; *Figure 6—figure supplement 2B*). The opposite is true for the HIF-1α responsive *PPFIA4* promoter and *PTPRN* enhancer (*Figure 6A*; *Figure 6—figure supplement 2A*): a specific increase in HIF-1α and HIF-2α/1α binding selectively induces both genes, which are instead insensitive to elevated HIF-2α and HIF-1α/2α levels.

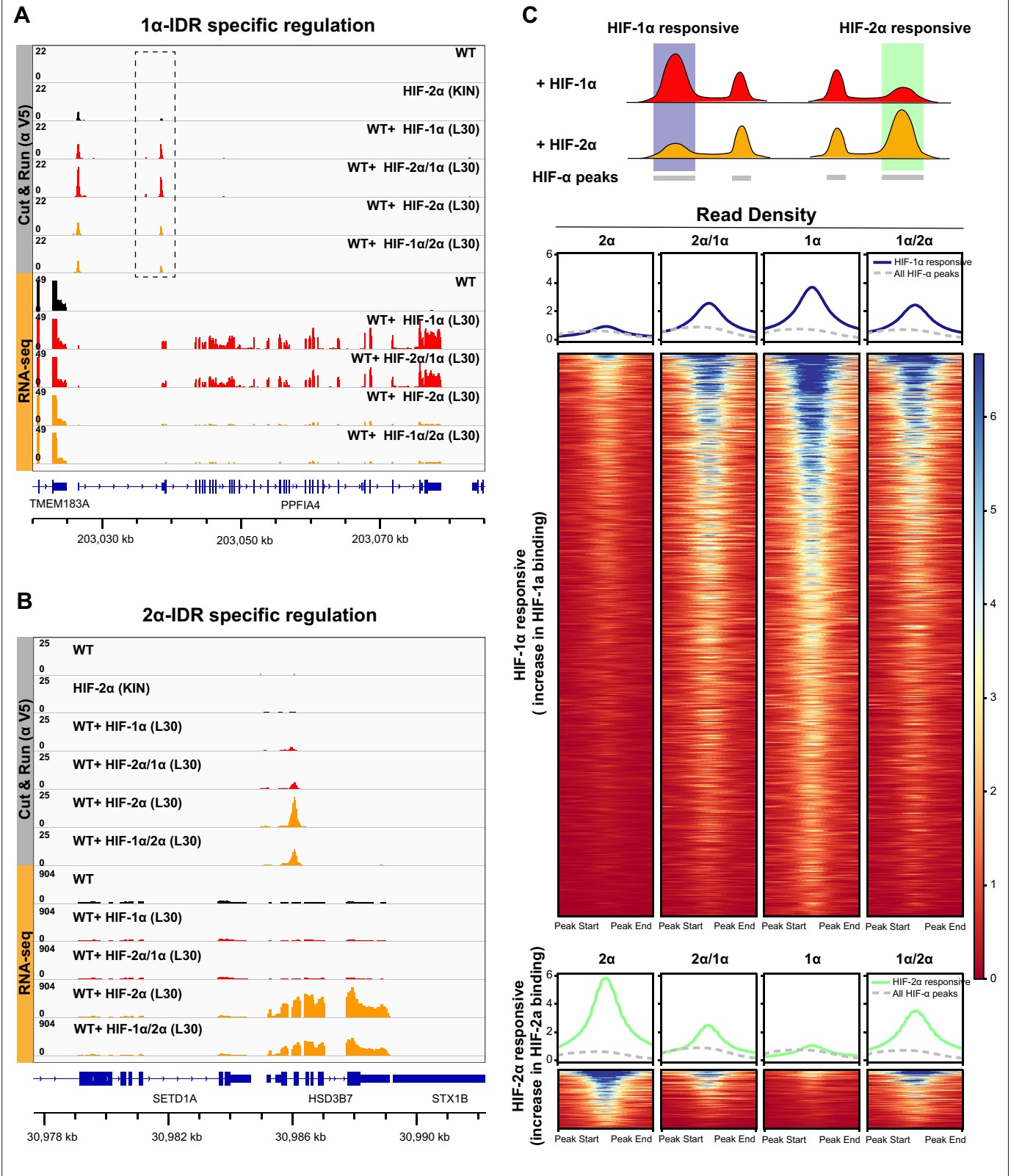

**Figure 6.** HIF-α intrinsically disordered region (IDR) contributes to isoform-specific chromatin binding preferences and activation of isoform-specific hypoxia-inducible factor (HIF) target genes. (**A–B**) Genome browser view of anti-V5 Cut&Run and RNA-seq results on wild-type (WT) cells, Halo-HIF-2α knock-in (KIN) cells and cells expressing different Halo-tagged HIF-α variants driven by an L30 promoter, showing IDR-specific regulation. (**A**) An example of genes that are preferentially bound (dashed box) and specifically activated by HIF-1α, as well as by HIF-2α/1α. (**B**) An example of genes

*Figure 6 continued on next page*

*Figure 6 continued*

that are preferentially bound and specifically activated by HIF-2α, as well as by HIF-1α/2α. (**C**) Genome-wide analysis of contribution of DNA-binding domain (DBD) or IDR on isoform-specific binding with Cut&Run data. Top: Schematic showing definition of HIF-1α and HIF-2α responsive regions. Regions that show increased binding when overexpressing Halo-tagged HIF-1α compared to when overexpressing Halo-tagged HIF-2α are defined as HIF-1α responsive regions (purple highlight). Regions that show increased binding when overexpressing Halo-tagged HIF-2α compared to when overexpressing HIF-1α are defined as HIF-2α responsive regions (green highlight). Bottom: Heatmap and pile-up results (blue curves for HIF-1α responsive regions and green curves for HIF-2α responsive regions) on binding strength at all HIF-1α responsive regions or HIF-2α responsive regions when cells overexpressing either Halo-tagged HIF-2α, -2α/1α, -1α, or -1α/2α. Pile-up enrichment results at all HIF binding sites (dashed gray curves) are used as control.

The online version of this article includes the following figure supplement(s) for figure 6:

**Figure supplement 1.** Cut&Run results show no difference in overall binding profiles between different HIF-α variants.

**Figure supplement 2.** Additional examples showing that HIF-α intrinsically disordered region (IDR) contributes to isoform-specific chromatin binding preferences.

**Figure supplement 3.** Results from analyzing Cut&Run individual replicates showing reproducibility.

These genome-wide analyses confirm that IDR-guided binding by HIF-2α versus HIF-1α can indeed differentially activate selected gene loci.

Because IDR-specific binding is not readily detectable at a genome-wide scale due to the intrinsic limitations of ensemble measurements, we next focused on differentially regulated sites only. We first identified sites that respond differentially to either HIF-1α or HIF-2α, and then ask what is the binding signal strength for the HIF-1α/2α and HIF-2α/1α chimeras – do they resemble more those of HIF-1α or HIF-2α, depending on their DBD or IDR (*Figure 6C*)? We define sites that show elevated binding only when HIF-1α is overexpressed as HIF-1α responsive (*Supplementary file 2*), and sites that show elevated binding only when HIF-2α is overexpressed as HIF-2α responsive (*Supplementary file 3*). For HIF-1α responsive regions, we observed elevated binding when overexpressing either HIF-1α/2α or HIF-2α/1α, compared to HIF-2α, indicating that HIF-1α DBD and IDR both contribute binding to HIF-1α responsive regions. For HIF-2α responsive regions, binding tended to be more elevated for HIF-1α/2α compared to HIF-2α/1α, suggesting that HIF-2α IDR might dominate binding to HIF-2α responsive sites (*Figure 6C*; *Figure 6—figure supplement 3*).

Unlike Cut&Run data, we observe clear genome-wide IDR-dependent gene regulation for RNA-seq performed on the same sets of cells. Specifically, cells overexpressing the HIF variants containing the same IDR show similar overall gene expression profiles (*Figure 7*). For example, genes activated by HIF-1α but not HIF-2α can also be activated by HIF-2α/1α, but not HIF-1α/2α. Overall, our genomic results show that HIF-α IDR contributes significantly to isoform-specific target site binding and helps determine isoform-specific target gene activation.

## Discussion

TFs must search, recognize, and bind to their specific target sites among millions of possible DNA sequences along chromatin to activate the correct gene. With the successful development of X-ray crystallography and cryo-EM, mechanisms of DNA-binding specificity have been extensively studied, primarily based on classically structured globular DBDs of TFs. We now know that a variety of structural mechanisms are used to recognize DNA, including formation of specific hydrogen bonds and DNA contour interactions (*Rohs et al., 2010*). However, these inherent binding modalities of DBDs alone cannot explain TF binding site selection in vivo in eukaryotic cells. As revealed by genome-wide in vivo binding assays, only a subset of potential target sites become occupied, and this is not entirely consistent with either DNA-binding site affinity or chromatin accessibility (*Behera et al., 2018*; *Grossman et al., 2017*; *Srivastava and Mahony, 2020*). On the other hand, TFs have long been recognized to also contain long unstructured transactivation domains with simple amino acid composition (Gln-rich, acidic, Pro-rich, etc.), which often posed challenges to purification and/or crystallization of full-length TFs (*Courey and Tjian, 1988*; *Ma and Ptashne, 1987*; *Mermod et al., 1989*; *Tjian and Maniatis, 1994*). Recently, such IDRs were reported to play an important role in weak and multivalent protein-protein interactions to form local small transient hubs that, when exacerbated by overexpression, can drive phase separation. Although not structurally defined, these interactions can still be sequence/amino acid composition selective (*Chong et al., 2018*; *Chong and Mir, 2021*).

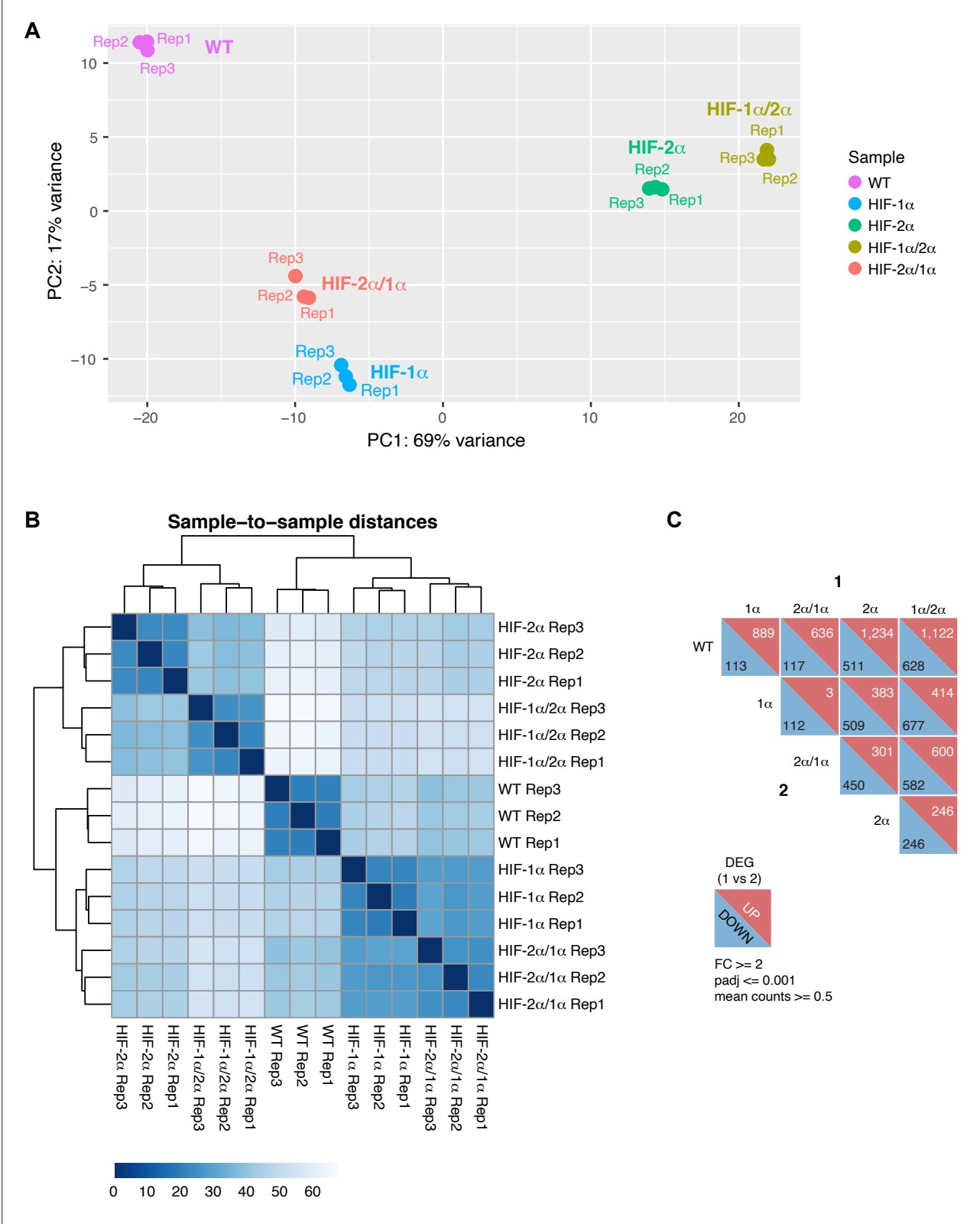

**Figure 7.** RNA-seq analysis shows HIF-α isoform-specific gene regulation is intrinsically disordered region (IDR)-dependent. (**A–B**) Principle component analysis (PCA) (**A**) and clustering (**B**) of RNA-seq results performed on wild-type (WT) 786-O cells, and cells expressing Halo-tagged HIF-α variants driven by an L30 promoter. HIF-α variants sharing the same IDR (-1α and -2α/1α or -2α and -1α/2α) have similar gene expression profiles. (**C**) DESeq2 output summarizing pair-wise comparison of the number of differentially expressed genes (DEG) in WT cells and cells overexpressing various HIF-α forms.

IDRs are now proposed to have important functions in boosting gene expression through hub or condensate formation to locally enrich for factors that are needed for transcription (*Boija et al., 2018*; *Cho et al., 2018*; *Chong et al., 2018*; *Sabari et al., 2018*; *Wei et al., 2020*). However, few studies of IDRs have investigated their potential role in DNA-binding site search and selection. Some studies reported that for a subset of zinc finger proteins (Sp2 and KLF3), an IDR is critical for in vivo binding and specificity (*Burdach et al., 2014*; *Lim et al., 2016*; *Völkel et al., 2015*), and two recent studies using genomic approaches reported the IDR as a determinant for specificity for some of the yeast TFs (*Brodsky et al., 2020*; *Gera et al., 2022*).

Here, using advanced live-cell SPT, we report that TF IDRs previously associated with ADs are, in fact, an important determinant mediating nuclear search dynamics and chromatin binding characteristics. Employing both genetic and small-molecule perturbations together with a series of domain-swap and mutation experiments, we found that it is the AD-associated disordered region of HIF-α rather than the intrinsic molecular weight of the TF that dictates a relatively slow diffusion for both HIF-α monomers and HIF-α/β dimers. On the other hand, when not engaged with HIF-α, HIF-1β

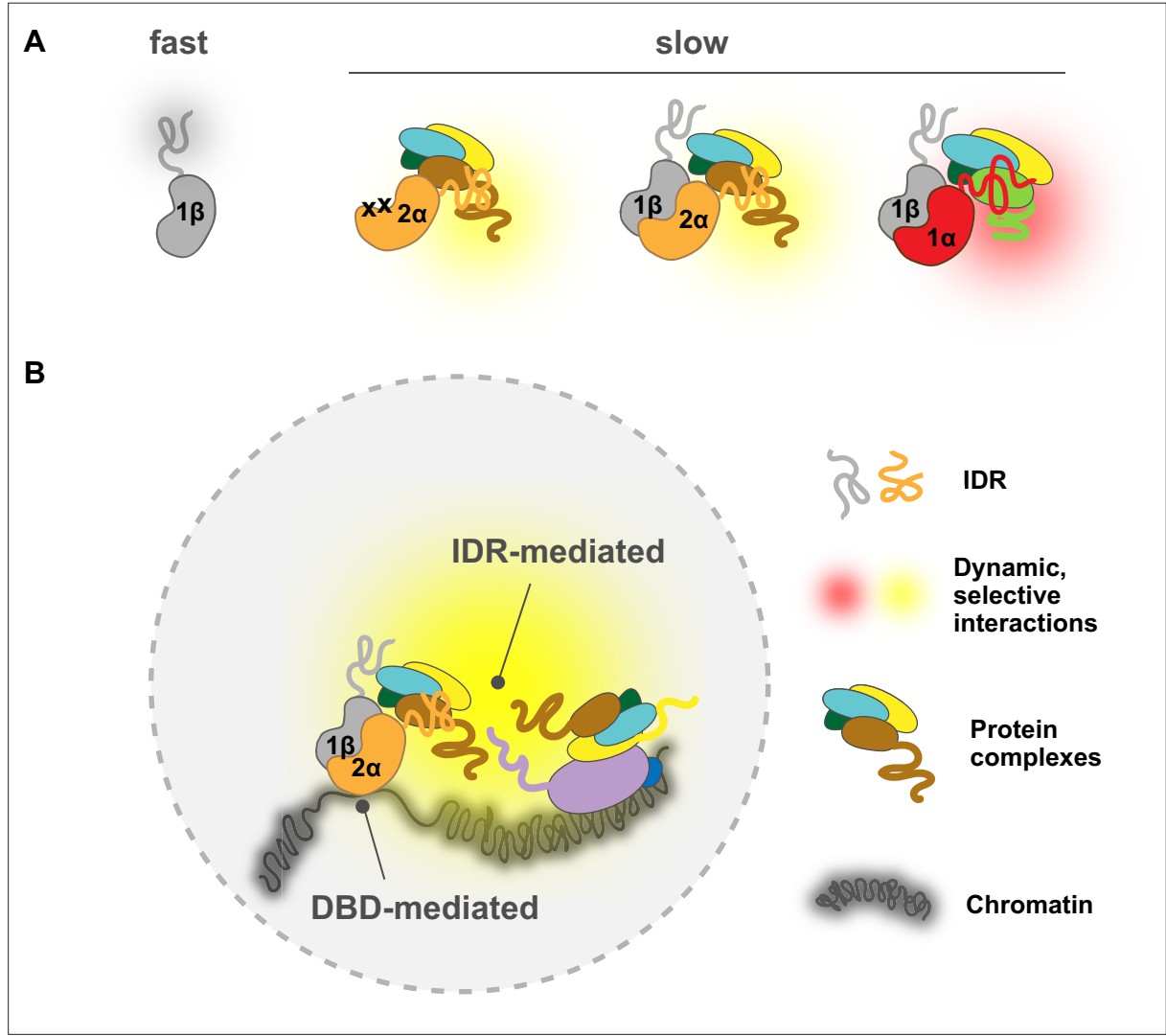

**Figure 8.** A model for intrinsically disordered region (IDR)-mediated nuclear search and chromatin binding. (**A**) The HIF-α IDR determines its slow motion of both the HIF-α monomer and HIF-α/β dimer, likely by HIF-α IDR-mediated interactions with nuclear macromolecules. For HIF-α, the IDR thus determines its slow motion regardless of its dimerization status. For HIF-1β, dimerization slows it down due to extra interactions (yellow and red clouds) brought by HIF-α IDR. (**B**) HIF-chromatin engagement comprises two components: DNA-binding domain (DBD)- and IDR-mediated interactions. As an obligated dimer, the DBD and the dimerization domain are both necessary for HIF binding, but the IDR determines the bound fraction, possibly via its interaction with nearby macromolecules, including other proteins and/or nucleic acids (DNA and/or RNA).

diffuses rapidly as expected for an unencumbered subunit. These results indicate that the diffusion characteristic of HIF molecules is profoundly influenced by the properties of their disordered regions (*Figure 8A*). In fact, computational analysis shows very different amino acid composition bias among HIF-1α, -2α, and -1β disordered regions (*Figure 1—figure supplement 1B*). Thus, it is very likely that as these molecules navigate through the crowded nuclear environment, their distinct stretches of IDRs that also contain ADs make differential and selective interactions with other nuclear components, resulting in distinct diffusive behaviors. According to Coulomb's law, acidity, and thus electrostatic charge on the molecules, contributes to the attraction or repulsion forces toward other molecules in the environment. Thus it is also possible that due to differences in acidity, the different charges on these IDRs can cause differential non-specific interactions with macromolecules including not only proteins, but also DNA and RNA (*Xiang et al., 2020*).

While it is easy to conceptualize how IDRs can influence the rate of diffusion, one unexpected result is that they also largely determine how much TFs bind to chromatin, as well as influence their binding site preferences. Although we confirmed that the DBD and dimerization domains are important for chromatin binding, the surprise was that our domain-swap experiments clearly demonstrated that the percentage of bound TF is mainly contributed by regions outside of the DBD/dimerization domains. One explanation could be the differential charge propensities of the different disordered regions (*Figure 1—figure supplement 1B*). For example, the HIF-2α IDR is more positively charged and may not only slow down nuclear exploration but also stabilize chromatin binding, possibly through stronger non-specific interactions with negatively charged chromatin-associated RNA and/or nucleosome-free DNA regions. Besides direct chromatin interactions, HIF-2α IDR could also increase and stabilize binding via indirect interactions with other chromatin-bound proteins. Moreover, since different IDRs can selectively interact with other IDRs (*Chong et al., 2018*; *Chong and Mir, 2021*), we also postulate that selective interactions with other TFs or co-regulators may play a role in determining HIF chromatin-binding specificity. One hint of such a 'combinatorial TF selectivity mechanism' is the enrichment of different TF motifs at HIF-1α and HIF-2α binding sites. Depending on the cell type, AP-1 or FOXD2, FOXL1 and FOXC2 were previously found at HIF-2α binding sites, while HEY1/2, ZNF263, or SP1/2 were found at HIF-1α binding sites (*Smythies et al., 2019*). It was also previously reported that while no target specificity was preserved in reporter gene assays, the N-terminal TAD of HIF-α conferred endogenous target specificity for two of the HIF-1 unique genes examined, possibly via specific interactions with transcriptional co-factors (*Hu et al., 2007*). Further Co-IP or pull-down assay coupled with mass spectrometry will be needed to more fully dissect this type of in vivo selectivity mechanism.

Given our results, we believe that the simple 'division of labor' model for DBDs and IDRs is an over-simplification which was probably only suitable for prokaryotes. Instead, eukaryotic TFs have evolved to exploit both DBDs and IDRs for chromatin binding as well as binding site selection, which is best suited for the eukaryotic chromatin environment. We propose that TF-chromatin engagement (binding and site selection) comprises two components – the first being DBD-mediated, mainly the classical motif binding force that acts directly on DNA, and the second being IDR-mediated interactions (likely weak and multivalent) with the native chromatin environment that include protein-protein interactions, and interactions with DNA and RNA molecules. Thus, we want to emphasize that the term 'DNA-binding specificity' refers to the first interaction component and we propose it should only be used with respect to isolated or naked DNA in eukaryotes. Instead, 'chromatin binding preferences' would be a better term to describe in vivo binding preferences of eukaryotic TFs within a native chromatin environment, which not only include DBD-DNA interactions, but also involve 'micro-environment recognition' mediated by IDR-chromatin interactions. These two forces can be additive or competitive, depending on the macromolecule composition of each specific chromatin locus, resulting in varied TF occupancy that could not be simply predicted from the DNA code. It seems that in the case of HIFs, for most of their binding sites the combined DBD- and IDR-mediated forces are similar for both HIF-1α and -2α, resulting in overlapping occupancy for different isoforms at these sites. However, for a few but striking cases, IDR-mediated interaction dominates HIFs' binding preferences, modulating the amount of HIF isoforms at these differentially regulated sites (*Figure 8B*). Interestingly, Myc, a closely related bHLH-LZ family TF, shows predominately DBD-mediated chromatin engagement, where non-specific DBD-DNA interactions contribute most of the binding (*Pellanda et al., 2021*). These different observations suggest that different eukaryotic TFs

may exploit DBD- and IDR-mediated interactions to different degrees, resulting in distinct chromatin engagement mechanisms.

We note that IDR-mediated chromatin engagement could happen either before and after DNA binding. It is possible that these IDR-mediated interactions first locate the TF to a few highly concentrated protein hubs in the vicinity of selected genomic binding sites, and then the TF performs a more localized target search with the DBD to find its motif, as it has been recently proposed (*Jana et al., 2021*; *Darzacq and Tjian, 2022*; *Staller, 2022*). In this case, whether the TF explores the nucleus alone to join the hubs or explores it as a larger complex containing other TFs/coactivators, IDRs both determine the TF dynamics and significantly improves the TF on-rate. On the other hand, it is also possible that TFs first binds to their target site via the DBD, and then IDR-mediated interactions stabilize the binding, increasing the TF's residence time and decreasing its off-rate. In both cases, distinct IDRs among different TF from closely related family members can create isoform-specific binding (and thus achieve isoform-specific gene activation) at selected genomic sites among all binding sites that are shared by the same family.

Finally, we have demonstrated that our fSPT platform provides a powerful tool able to resolve in vivo protein dynamics that is exquisitely sensitive to concentration, subunit stoichiometry, and genetic/small-molecule perturbations. This is especially important when studying TFs, where a slight difference in expression level often generates completely different results, rendering over-expression systems highly susceptible to artifacts. It is also worth underscoring the importance of studying TFs in their native physiologically relevant chromatin environment, given their obligate interactions with higher-order chromatin structures and co-factors. For example, the EPO gene is reported to be responsive to HIF-2α but not HIF-1α in Hep3B cells (*Warnecke et al., 2004*) and in murine liver (*Rankin et al., 2007*), however, a luciferase reporter driven by the upstream EPO enhancer also responds strongly to HIF-1α (*Varma and Cohen, 1997*), which may generate misleading results and interpretations. Our fSPT platform allows us to study transcriptional regulation in the native chromatin context and with endogenous TF levels to obtain data with physiological and functional relevance. Such live-cell real-time measurements under native cell contexts could prove to be highly valuable, both for dissecting in vivo mechanisms of transcription regulation and for guiding the development of effective therapeutics. Our Belzutifan treatment experiment is an example of how fSPT can reveal the mechanism of action of small-molecule inhibitors, and how it could serve as a powerful tool to screen for drugs that selectively target one isoform versus another, using dimerization and binding readouts as indicators of efficacy and specificity. Moreover, since our results demonstrated how IDRs can affect TF diffusion behavior, potentially distinct dynamic features determined by a particular IDR can be exploited as a readout for screening small molecules or peptides that target allosteric sites of TFs. Assays that can quantitatively measure TF diffusive behavior in live cells could be transformative for advancing drug discovery because a high-throughput imaging strategy opens the door to effectively target what has been traditionally considered 'undruggable', such as most protein-protein interactions including potentially unstructured TF ADs.

In summary, using the HIF protein family as a case study, we uncovered a mechanism of IDR-mediated nuclear search and differential chromatin binding leading to selective gene activation. We expect this fundamental principle to be applicable to a broad range of TF families.

## Materials and methods
### Cell culture, stable cell line construction, and drug treatment

Human 786-O clear cell renal carcinoma cells were obtained from the UCB Cell Culture Facility (RRID:SCR_017924), which were originally obtained from ATCC (#CRL-1932). The cells were tested for mycoplasma using nuclear stain with a 100× lens on a fluorescence scope on Janauary 22, 2020 when banked and STR was used to identify them on January 24, 2020. Cells were cultured at 37°C with 5% $CO_2$ in 4.5 g/l glucose DMEM (Thermo Fisher, Waltham, MA, #10566016) supplemented with 10% fetal bovine serum (HyClone, Logan, UT, Cat. #SH30396.03, lot #AE28209315), 1 mM sodium pyruvate (Thermo Fisher #11360070), and 100 U/mL penicillin-streptomycin (Thermo Fisher #15140122). Cells were subcultured at a ratio of 1:4 to 1:12 every 2–4 days for no longer than 30 days. A regular mycoplasma testing was performed every 2 weeks using PCR. Phenol red-free DMEM (Thermo Fisher,

**Table 1.** Constructs used to generate stable cell lines.

| Name | Promoter | Gene product | Short name in the paper | Appeared in |
|---|---|---|---|---|
| PB EF1a 3XF EX-MCS IRES Puro | EF1a | 3xFLAG tag | 3xF | *Figure 3* |
| PB EF1a 3XF-GDGAGLIN-hEPAS1 IRES Puro | EF1a | HIF-2α N-terminally fused with 3xFLAG tag through a short peptide linker sequence (GDGAGLIN) | HIF-2α | *Figure 2* *Figure 3* *Figure 2—figure supplement 2* |
| PB EF1a 3XF-GDGAGLIN-hEPAS1_R171A-V192D IRES Puro | EF1a | HIF-2α dimerization mutant (R171A-V192D) N-terminally fused with 3xFLAG tag through a short peptide linker sequence (GDGAGLIN) | HIF-2α DM | *Figure 2* *Figure 2—figure supplement 2* |
| PB EF1a 3XF-EPAS1_365 IRES Puro | EF1a | The N terminal region of HIF-2α (aa 1–365) N-terminally fused with 3xFLAG tag | HIF-2α NT | *Figure 3—figure supplement 2* |
| PB EF1a 3XF-EPAS1_365–364_HIF1a IRES Puro | EF1a | HIF-2α/1α chimera protein (aa 1–365 of HIF-2α and aa 364–826 of HIF-1α) N-terminally fused with 3xFLAG tag | HIF-2α/1α | *Figure 3* |
| PB EF1a 3XF-GDGAGLIN-hHIF1a IRES Puro | EF1a | HIF-1α N-terminally fused with 3xFLAG tag through a short peptide linker sequence (GDGAGLIN) | HIF-1α | *Figure 3* |
| PB EF1a 3XF-HIF1a_363 IRES Puro | EF1a | The N terminal region of HIF-1α (aa 1–363) N-terminally fused with 3xFLAG tag | HIF-1α NT | *Figure 3—figure supplement 2* |
| PB EF1a 3XF-HIF1a_363–366_EPAS1 IRES Puro | EF1a | HIF-1α/2α chimera protein (aa 1–363 of HIF-1α and aa 366–870 of HIF-2α) N-terminally fused with 3xFLAG tag | HIF-1α/2α | *Figure 3* |
| PB L30prom V5-Halo-GDGAGLIN-hEPAS1 IRES Puro | L30 | HIF-2α N-terminally fused with V5-HaloTag through a short peptide linker sequence (GDGAGLIN) | Halo-HIF-2α (L30) | *Figure 4* *Figure 4—figure supplement 2* |
| PB L30prom V5-Halo-GDGAGLIN-hEPAS1_365–364_HIF1a IRES Puro | L30 | HIF-2α/1α chimera protein (aa 1–365 of HIF-2α and aa 364–826 of HIF-1α) N-terminally fused with V5-HaloTag through a short peptide linker sequence (GDGAGLIN) | Halo-HIF-2α/1α (L30) | *Figure 4* |
| PB L30prom V5-Halo-GDGAGLIN-hHIF1a IRES Puro | L30 | HIF-1α N-terminally fused with V5-HaloTag through a short peptide linker sequence (GDGAGLIN) | Halo-HIF-1α (L30) | *Figure 4* |
| I_PB L30prom V5-Halo-GDGAGLIN-HIF1A_363–366_EPAS1 IRES Puro | L30 | HIF-1α/2α chimera protein (aa 1–363 of HIF-1α and aa 366–870 of HIF-2α) N-terminally fused with V5-HaloTag through a short peptide linker sequence (GDGAGLIN) | Halo-HIF-1α/2α (L30) | *Figure 4* |
| PB L30prom V5-Halo-GDGAGLIN-hEPAS1 IRES Puro_R27E | L30 | HIF-2α DBD mutant (R27E) N-terminally fused with V5-HaloTag through a short peptide linker sequence (GDGAGLIN) | Halo-HIF-2α, R27E (L30) | *Figure 5* |
| PB L30prom V5-Halo-GDGAGLIN-hHIF1a IRES Puro_R30E | L30 | HIF-1α DBD mutant (R30E) N-terminally fused with V5-HaloTag through a short peptide linker sequence (GDGAGLIN) | Halo-HIF-1α, R30E (L30) | *Figure 5* |
| PB L30prom V5-Halo-GDGAGLIN-hEPAS1_R171A-V192D IRES Puro | L30 | HIF-2α dimerization mutant (R171A-V192D) N-terminally fused with V5-HaloTag through a short peptide linker sequence (GDGAGLIN) | Halo-HIF-2α DM (L30) | *Figure 5* |
| PB L30prom V5-Halo-GDGAGLIN-hHIF1a_R170A-V191D IRES Puro | L30 | HIF-1α dimerization mutant (R170A-V191D) N-terminally fused with V5-HaloTag through a short peptide linker sequence (GDGAGLIN) | Halo-HIF-1α DM (L30) | *Figure 5* |

#21063029) supplemented with 10% fetal bovine serum, 1 mM sodium pyruvate, and 100 U/mL penicillin-streptomycin was used for imaging.

Stable cell lines expressing the exogenous gene product (*Table 1*) were generated by PiggyBac transposition and antibiotic selection. The gene of interest was cloned into a PiggyBac vector which also co-expresses a puromycin resistant gene using Gibson Assembly and confirmed by Sanger sequencing. Cells were transfected by nucleofection using the Lonza Cell Line Nucleofector Kit V (Lonza, Basel, Switzerland, #VVCA-1003) and the Amaxa Nucleofector II device. For each transfection, cells were plated 1–2 days before nucleofection in a 15 cm dish, and reached approximately 50–70% confluency on the day of nucleofection, which equals to approximately 3–4 million cells. Two µg of PiggyBac plasmid was co-transfected with 1 µg of SuperPiggyBac transposase vector with the T-020 program according to the manufacturer's protocol. Transfected cells were cultured for 24–48 hr before changing to selection media. Cells were then selected for 14 days with 1 µg/ml puromycin (Thermo Fisher #A1113803) and stable cell lines were maintained in selection media for up to 30 days of culturing.

For drug treatment, 100 mM Belzutifan stock solution was prepared by dissolving Belzutifan powder (CAS No: 1672668-24-4, MedChemExpress, Monmouth Junction, NJ, Cat. #HY-125840) in DMSO (Sigma, St Louis, MO, #D2650), and was diluted 1:500,000 in growth media to the final concentration of 0.2 µM. The same volume of DMSO (0.0002%) is used in the reference group as control. Cells were treated for 24 hr in either Belzutifan or DMSO alone before imaging. For dosage-dependent assays in *Figure 3—figure supplement 1*, DMSO amount was kept the same (0.0002%) for all drug concentrations.

## CRISPR/Cas9-mediated genome editing

KIN cell lines were generated as previously described (*Hansen et al., 2017*) with the following changes. For each editing case, we designed three sgRNAs using CRISPOR (*Concordet and Haeussler, 2018*). For each guide/donor pair, approximately 4 million 786-O cells were nucleofected with 3.75 µg of donor plasmid and 1.25 µg of sgRNA plasmid. Twenty-four hr after transfection, Venus-positive cells were sorted and cultured for another 5–7 days, then Halo-positive cells were sorted individually into single wells of 96-well plates. Clones were expanded and genotyped with two rounds of PCR. The first round used one primer upstream of the left homologous arm and the other primer downstream of the right homologous arm. The second round used either of the external primers and a corresponding internal primer located in the HaloTag coding region. Homozygous clones with the correct genotype, including Halo-HIF-2α KIN clone A31 and clone B50, Halo-HIF-1β KIN clone A21 and clone B89, were confirmed by Sanger sequencing and western blotting.

## Cell preparation and dye labeling for imaging

For fSPT, cells were grown on sonicated and plasma-cleaned 25 mm circular No. 1.5H precision cover glass (Marienfeld, Germany, 0117650) in six-well plate. At least 1 day before imaging, selective medium (if used) was removed and replaced with non-selective growth medium. On the day of imaging, cells should be less than 100% confluent. Immediately before imaging, cells were double-labeled with JFX dyes as follows: cells were first incubated for 5 min in 1 ml growth medium containing JFX646, at a concentration that only gives approximately 10 detected molecules per frame in the initial frames to ensure minimum misconnection of trajectories between detections. This concentration differs from cell line to cell line, ranging from 0.2 to 5 nM, depending on the expression level of the Halo-fusion protein. After 5 min of incubation, medium was removed, cells were rinsed in PBS, and incubated for 5 min in 1 ml medium containing JFX549. The concentration of JFX549 also varies, usually at 25× the concentration of JFX646. After incubation, cells were washed twice for 5 min each, a first time with 2 ml regular growth media, and a second time with 2 ml phenol red-free growth media, with a quick PBS rinse before each wash. After wash, coverslip was transferred to Attofluor Cell Chambers (Thermo Fisher, #A7816) with cells facing up and 1 ml phenol red-free medium added to the chamber. For Belzutifan treatment experiments, Belzutifan or equivalent amount of DMSO was added throughout the labeling and washing steps (except during PBS rinses), as well as in the final imaging medium, at the indicated concentration.

## Live-cell SPT

All SPT experiments were carried out on a custom-built microscope as previously described (*Hansen et al., 2017*; *McSwiggen et al., 2019*). In brief, a Nikon TI microscope is equipped with a 100×/NA

1.49 oil-immersion TIRF objective, a motorized mirror, a perfect Focus system, an EM-CCD camera, and an incubation chamber maintained with humidified atmosphere with 5% $CO_2$ at 37°C. All microscope, camera, and hardware components were controlled through the NIS-Elements software (Nikon).

During imaging, samples were excited with 561 nm laser at 1100 mW (Genesis Coherent, Santa Clara, CA) with emission filter set to Semrock 593/40 nm bandpass filter to locate and focus the cell nuclei, as well as to adjust laser angle to achieve highly inclined laminated optical sheet (HiLo) illumination (*Tokunaga et al., 2008*). An ROI (region of interest) of random size was selected to fit into the interior of the nuclei but with maximized area. Then the emission filter was switched to Semrock 676/37 nm bandpass filter while keeping TIRF angle, stage xyz position, and ROI the same. Because we achieved sparse labeling using low dye concentrations, pre-bleaching was not required to visualize single molecules. Movies were then taken with 633 nm laser (Genesis Coherent, Santa Clara, CA) at 1100 mW and 1 ms pulse, with camera exposure at 5.48 ms frame rate for 800–1600 frames, until samples were completely photobleached. At least 20 movies (corresponding to 20 cells) were taken for each sample as one biological replicate on a given day. A total of three biological replicates on three separate days were collected to produce the final results (>60 cells per cell line/condition).

## SPT data processing

Raw SPT movies were processed with an earlier version of saspt package for state array (SA) method (*Heckert et al., 2022*), which is publicly available at https://github.com/alecheckert/quot (*Heckert, 2019*) to generate trajectory files (.trajs). Generally, it performs tracking in the following steps: read a frame, find spots in the frame, localize spots to subpixel resolution, and reconnect spots from consecutive frames into trajectories. Since a non-photoactivatable dye was used for all SPT experiments, we labeled cells with a dye concentration that only gives very low spot detection density, which allowed us to track spots from the very first frame. This is important because if the initial frames are filtered due to high localization density, there might be a bias toward moving molecules, due to the bound molecules being photobleached and diffusing molecules moving into the focal plane during the later frames. Although we used very sparse labeling, occasionally there would be frames with high density, to minimize misconnections due to multiple particles in close proximity, we incorporated a filtering step where we removed frames with more than seven detections in the following way. First, we computed the number of detections per frame. Next, this function was smoothed with uniform filtering with a kernel width of 21 frames. Finally, we identified frames with fewer than seven detections after smoothing and isolated trajectories from these frames. Specifically, the following configuration was used for all detections and tracking: Image reading and filtering settings: start = 0, method = 'identity', chunk_size = 100; Spot detection settings: method = 'llr', k=1.0, w=15, t=18; Subpixel localization settings: method = 'ls_int_gaussian', window_size = 9, sigma = 1.0, ridge = 0.001, max_iter = 20, damp = 0.3; Tracking settings: method = 'euclidean', max_spots_per_frame = 7, pixel_size_um = 0.16, frame_interval = 0.00548, search_radius = 1.0, max_blinks = 0, min_I0=0.0, scale = 7.0.

To infer the distribution of diffusion coefficients from experimentally observed trajectories, we used an earlier version of saspt package for state array (SA) method (*Heckert et al., 2022*), which is publicly available at https://github.com/alecheckert/spagl (*Heckert, 2020*) (sample_script_fss.py), which generates the posterior mean occupations for an SA evaluated on trajectories across all cells. In all analyses, we used the likelihood function for RBME (*Heckert et al., 2022*). Settings were: frame_interval = 0.00548, pixel_size_um = 0.16, dz = 0.7. Occupations are reported as the mean of the posterior distribution over state occupations, marginalized on diffusion coefficient.

To generate RBME likelihood for individual cells (*Heckert et al., 2022*), we used the sample_script_by_file.py script in the same repository (https://github.com/alecheckert/spagl) (*Heckert, 2020*) with the following settings: frame_interval = 0.00548, dz = 0.7, pixel_size_um = 0.16, scale_by_total_track_count = True, scale_colors_by_group = True.

## Antibodies

The following antibodies were used for ChIP-seq: rabbit polyclonal anti-HIF-2α (Novus Biologicals, Centennial, CO, #NB100-122), mouse monoclonal anti-HIF-1β (Novus Biologicals, #NB100-124), rabbit polyclonal anti-V5 (Abcam, Cambridge, UK, #ab9116). The following antibodies were used for Cut&Run: mouse monoclonal anti-V5 tag (Thermo Fisher, #R960-25) diluted to 0.01 mg/ml, mouse IgG (Jackson ImmunoResearch #015-000-003) diluted to 0.01 mg/ml, rabbit anti-mouse IgG H&L

(Abcam, #ab46540) diluted to 0.01 mg/ml. The following antibodies were used for western blotting: rabbit monoclonal anti-HIF-2α (Cell Signaling, Danvers, MA, #D9E3) diluted at 1:1000, rabbit monoclonal anti-HIF-1β (Cell Signaling, #D28F3) diluted at 1:1000, mouse monoclonal anti-V5 tag (Thermo Fisher, #R960-25) diluted at 1:2500, mouse monoclonal anti-HaloTag (Promega, Madison, WI, #G9211) diluted at 1:1000, mouse monoclonal anti-TBP (Abcam, #ab51841) diluted at 1:2500, goat-anti-mouse-HRP (Thermo Fisher, #31430) diluted at 1:2000, goat-anti-rabbit-HRP (Thermo Fisher, #31462) diluted at 1:2000.

## Western blotting

All western samples were prepared as follows: cells growing in either six-well plates or 10 cm dish in log phase were rinsed with PBS twice and lysed on ice in 100–500 µl 2× sample buffer (80 mM Tris pH 6.8, 2% SDS, 10% glycerol, 0.0006% bromophenol blue) containing 280 mM 2-mercaptoethanol (Sigma #M7522), 1× aprotinin (Sigma, #A6279, diluted 1:1000), 1 mM benzamidine (Sigma, #B6506), 1× cOmplete EDTA-free Protease Inhibitor Cocktail (Sigma, #5056489001), and 0.25 mM PMSF (Sigma #11359061001). Cell lysates were scraped and collected into 1.5 ml Eppendorf tubes, incubated at 99°C with constant shaking, snap-frozen in liquid nitrogen, and stored at –80°C. On the day of western blotting, samples were thawed and centrifuged at top speed for 5 min at 4°C. Ten to 15 µl supernatant were loaded on an 8% SDS-PAGE gel, ran for 1 hr at 200 V and 4°C, and transferred to 0.45 µm nitrocellulose membrane (Thermo Fisher, #45004031) for 2 hr at 100 V. Membranes were blocked in 10% milk in 0.1% TBS-Tween for 1 hr at room temperature (RT), and incubated overnight at 4°C with primary antibodies diluted in 5% milk in 0.1% TBS-Tween. After 4×5 min washes in 0.1% TBS-Tween, membranes were incubated at RT for at least 1 hr with secondary antibodies diluted in 5% milk in 0.1% TBS-Tween. After 4×5 min washes in 0.1% TBS-Tween, membranes were incubated for 3 min in freshly made Perkin Elmer LLC Western Lightning Plus-ECL, Enhanced Chemiluminescence Substrate (Thermo Fisher, #509049326), and imaged with a Bio-Rad ChemiDoc imaging system (Bio-Rad, Model No: Universal Hood III). For reblotting, membranes were immersed in Restore Western Blot Stripping Buffer (Thermo Fisher, #21059) for 15 min at RT with shaking, washed 3×10 min in 0.1% TBS-Tween, followed by blocking, antibody incubation, and chemiluminescence reaction as described above.

## Luciferase reporter assay

The firefly luciferase reporter gene construct was made by inserting a 3× HREs from the EPO gene enhancer (sequence: tcgaagccctacgtgctgtctcacacagcctgtctgacctctcgacctaccggccgttcgaagccctacgtg ctgtctcacacagccttctgatctcgacctaccggccgttcgaagccctacgtgctgtctcacacagcctgtctgacctctcgacctaccg-gccgt) into the 5′ of the minimal TATA-box promoter in the pGL4.23 [luc2/minP] vector (Promega #E841A). A control pHRL-TK vector (Promega #E2241) expressing Renilla luciferase with an HSV TK promoter was used as reference to normalize luciferase activity. Cells were co-transfected with 1 µg of firefly Luciferase vector and 0.1 µg Renilla luciferase vector by nucleofection with Lonza Cell Line Nucleofector Kit V (Lonza, #VVCA-1003) and the T-020 program in the Amaxa Nucleofector II device. After nucleofection, cells were resuspended in complete growth medium, and plated into 12-well plates with Belzutifan added to various concentrations as indicated. Twenty-four hr after nucleofection, cells were lysed and luciferase activity was analyzed with Dual-luciferase Reporter Assay System (Promega, #E1960) according to the manufacturer's protocol. The relative luciferase activity was calculated by normalizing firefly luciferase activity to the Renilla luciferase activity to control for transfection efficiency.

## Chromatin immunoprecipitation and ChIP-seq library preparation

Chromatin immunoprecipitation (ChIP) was performed as described with few modifications (*Testa et al., 2005*). WT 786-O or endogenously tagged KIN clones A31 (V5-Halo-HIF-2α) and A21 (V5-Halo-HIF-1β) were expanded to two 15 cm dishes and cross-linked 5′ at RT with 1% formaldehyde-containing FBS-free medium; cross-linking was stopped by adding PBS-glycine (0.125 M final). Cells were washed twice with ice-cold PBS, scraped, centrifuged for 10′, and pellets were flash-frozen. Cell pellets were thawed and resuspended in 2 ml of cell lysis buffer (5 mM PIPES, pH 8.0, 85 mM KCl, and 0.5% NP-40, 1 ml/15 cm plate) w/ protease inhibitors and incubated for 10′ on ice. Lysates were centrifuged for 10′ at 4000 rpm and nuclear pellets resuspended in 6 volumes of sonication buffer (50 mM Tris-HCl, pH 8.1, 10 mM EDTA, 0.1% SDS) w/ protease inhibitors, incubated on ice for 10′, and sonicated to obtain

DNA fragments around 500 bp in length (Covaris S220 sonicator, 20% Duty factor, 200 cycles/burst, 150 peak incident power, 10 cycles 30" on and 30" off). Sonicated lysates were cleared by centrifugation and chromatin (400 μg per antibody) was diluted in RIPA buffer (10 mM Tris-HCl, pH 8.0, 1 mM EDTA, 0.5 mM EGTA, 1% Triton X-100, 0.1% SDS, 0.1% Na-deoxycholate, 140 mM NaCl) w/ protease inhibitors to a final concentration of 0.8 μg/μl, precleared with Protein G sepharose (GE Healthcare) for 2 hr at 4°C and immunoprecipitated overnight with 4 μg of specific antibodies. About 4% of the precleared chromatin was saved as input. Immunoprecipitated DNA was purified with the Qiagen QIAquick PCR Purification Kit, eluted in 33 μl of 0.1× TE (1 mM Tris-HCl pH 8.0, 0.01 mM EDTA) and analyzed by qPCR together with 2% of the input chromatin prior to ChIP-seq library preparation (SYBR Select Master Mix for CFX, Thermo Fisher). ChIP-qPCR primer sequences were as follows:

> hWISP1_positive_forward: TGAGGTCAGTGTGGTTTGGT.
> hWISP1_positive_reverse: ACATGGTCACGTAGCTAGCA.
> hWISP1_negative_forward: AGTCCCCAGCACATAGAAGG.
> hWISP1_negative_reverse: GGTTCTGAAGGTGACCGACT.

ChIP-seq libraries were prepared using the NEBNext Ultra II DNA Library Prep Kit for Illumina (NEB E7645) according to the manufacturer's instructions with a few modifications. Twenty ng of ChIP input DNA (as measured by Nanodrop) and 25 μl of the immunoprecipitated DNA were used as a starting material and the recommended reagents' volumes were cut in half. The NEBNext Adaptor for Illumina was diluted 1:10 in Tris/NaCl, pH 8.0 (10 mM Tris-HCl pH 8.0, 10 mM NaCl) and the ligation step extended to 30'. After ligation, a single purification step with 0.9× volumes of Agencourt AMPure XP PCR purification beads (Beckman Coulter A63880) was performed, eluting DNA in 22 μl of 10 mM Tris-HCl pH 8.0. Twenty μl of the eluted DNA were used for the library enrichment step, performed with the KAPA HotStart PCR kit (Roche Diagnostics KK2502) in 50 μl of total reaction volume (10 μl 5× KAPA buffer, 1.5 μl 10 mM dNTPs, 0.5 μl 10 μM NEB Universal PCR primer, 0.5 μl 10 μM NEB index primer, 1 μl KAPA polymerase, 16.5 μl nuclease-free water, and 20 μl sample). Samples were enriched with 9 PCR cycles (98°C, 45"; [98°C, 15"; 60°C, 10"] × 9; 72°C, 1'; 4°C, hold), purified with 0.9 volumes of AMPure XP PCR purification beads and eluted with 33 μl of 10 mM Tris-HCl pH 8.0. Library concentration, quality, and fragment size were assessed by Qubit fluorometric quantification (Qubit dsDNA HS Assay Kit, Invitrogen Q32851) qPCR and Fragment analyzer. Twelve multiplexed libraries (input, HIF-1β, HIF-1α, and V5 pull-downs in WT 786-O cells and A31 and A21 clones) were pooled and sequenced in one lane on the Illumina HiSeq4000 sequencing platform (50 bp, single end reads) at the Vincent J. Coates Genomics Sequencing Laboratory at UC Berkeley.

## ChIP-seq analysis

ChIP-seq raw reads from WT 786-O cells and A31 and A21 endogenously Halo-tagged clones (12 libraries total, 1 replicate per condition) were quality-checked with FastQC 0.10.1 and aligned onto the human genome (hg38 assembly) using Bowtie (*Langmead et al., 2009*), allowing for two mismatches (`-n 2`) and no multiple alignments (`-m 1`). Peaks were called with MACS2 2.1.0.20140616 (`--nomodel --extsize 300`) (*Zhang et al., 2008*) using input DNA as a control. To create heatmaps we used deepTools 2.4.1 (*Ramírez et al., 2016*). We first ran bamCoverage (`--binSize 50 --normalizeTo1x 2913022398 --extendReads 300 --ignoreDuplicates -of bigwig`) and normalized read numbers to 1× sequencing depth, obtaining read coverage per 50bp bins across the whole genome (bigWig files). We then used the bigWig files to compute read numbers across 6 kb centered on HIF-2α peaks called by MACS2 across all 786-O cell lines, subtracted of V5 peaks called by MACS2 in WT 786-O cells (computeMatrix reference-point `--referencePoint=TSS --upstream 3000 --downstream 3000 --missingDataAsZero --sortRegions=no`). We sorted the output matrices by decreasing WT 786-O enrichment, calculated as the total number of reads within a MACS2 called ChIP-seq peak. Finally, heatmaps were created with the plotHeatmap tool (`--averageTypeSummaryPlot=mean --colorMap='Blues' --sortRegions=no`).

## Cut&Run

Cut&Run was performed as published (*Janssens and Henikoff, 2019*) with the following modifications/specifications. Around 0.5 million cells were used for each experimental condition. Before permeabilization, proteinase inhibitor was left out from the buffer to minimize toxicity to the cells, but

was added during the permeabilization step. Digitonin was used at a final concentration of 0.02% for Dig-wash buffer and antibody buffer, as it was tested to be the minimum concentration to fully permeabilize 786-O cells. For each cell line, a primary antibody (either mouse-anti-V5 or mouse IgG, 1 mg/ml at 1:100 dilution) was used with a 4°C overnight incubation. A secondary rabbit-anti-mouse IgG was used at 1:100 dilution with an hour incubation at 4°C. For each wash, 100 μl of Dig-wash buffer was used. Chromatin digestion was done for 30 min on ice. DNA was extracted with phenol/chloroform and the pellet was dissolved in 30 μl 0.1× TE (1 mM Tris-HCl pH 8, 0.1 mM EDTA). DNA was quantified by Qubit (Qubit dsDNA HS Assay Kit, Invitrogen Q32851) and up to 40 ng (and up to 25 μl) was used for library preparation, using NEBNext Ultra II DNA Library Prep Kit for Illumina (NEB E7645) according to the manufacturer's instructions, with a few modifications. Reagents' volumes were cut in half. For end prep 20°C 30 min followed by 50°C 60 min was used. For adapter ligation, NEBNext Adapter was diluted 1:20 in Tris/NaCl, pH 8.0 (10 mM Tris, 10 mM NaCl), mixture was incubated for 30 min at 20°C (instead of 15 min). After ligation, product was size-selected with 1.75× Agencourt AMPure XP PCR purification beads (Beckman Coulter A63880), and DNA was eluted in 15 μl 10 mM Tris-HCl pH 8.0. 13 μl of eluted DNA was used for library enrichment with 15 μl NEBNext Ultra II Q5 Master mix, 1 μl 10 μM NEB Universal PCR primer and 1 μl 10 μM NEB Index Primer. Samples were enriched with 11 PCR cycles (98°C, 30"; [98°C, 10"; 65°C, 10"] × 11; 65°C, 5'; 4°C, hold). The 30 μl PCR product was size-selected with double AMPure XP cleanup (first with 24 μl [0.8×] beads and take the supernatant, then add 12 μl beads [to final PEG/NaCl 1.2×] and discard supernatant). Library was eluted with 15 μl 10 mM Tris-HCl pH 8.0 and quantified by Qubit (Qubit dsDNA HS Assay Kit, Invitrogen Q32851). Libraries were sent to MedGenome Inc (Foster City, CA) for fragment analysis, multiplexing and sequencing on the Illumina NovaSeq 6000 platform (150 bp, paired end reads).

## Cut&Run data analysis

Cut&Run raw fastq reads were first quality-checked with FastQC (http://www.bioinformatics.babraham.ac.uk/projects/fastqc) and then aligned to the human genome (hg38) using bowtie2 version 2.3.4.1 with options: `--local --no-unal --very-sensitive --no-mixed --no-discordant --phred33 -I 10 -X 700`. Non-human chromosomes were removed, then bam files were created using Samtools (*Li et al., 2009*) version 1.8. Then Sambamba (*Tarasov et al., 2015*) version 0.6.6 was used to sort bam files, and filter unmapped reads. Finally, samtools were used to index the files. For peak calling with MACS2 (`--keep-dup all --max-gap` 400 `-g hs --bdg -q` 0.01 `-f BAMPE`) (*Zhang et al., 2008*), mouse IgG control was used as control for each cell line. Blacklisted regions were removed from the output narrowpeak file with BEDTools (*Quinlan and Hall, 2010*) version 2.28.0. For visualization of genome-wide binding strength on the hg38 genome with the Integrative Genomics Viewer (IGV) (*Robinson et al., 2011*; *Thorvaldsdóttir et al., 2013*), individual replicates were first combined into a single bam file using Samtools version 1.8, and then converted to bigWig output files from deepTools bamCoverage (`--binSize` 20 `--normalizeUsing BPM --smoothLength` 60). The PCA and clustering analysis, as well as identification of differentially bound regions by HIF-1α and HIF-2α were done with DiffBind (*Stark and Brown, 2011*) version 3.0, with all three replicates as samples. The final HIF-1α and HIF-2α responsive regions were obtained by filtering the list with p-value <0.01. Heatmaps for binding strength were generated for each individual replicate or the combined replicates, using deepTools (*Ramírez et al., 2016*) version 3.5.1.

## RNA extraction and poly-A RNA-seq library preparation

Poly-A RNA-seq was performed in two batches and three biological replicates per sample. In the first batch (3 replicates each of WT 786-O cells, HIF-1β KIN, HIF-2α KIN, HIF-1α O.E., HIF-1α R30E, HIF-1α DM, HIF-2α O.E., HIF-2α R27E, HIF-2α DM) total RNA was extracted with the QIAGEN RNeasy Plus Mini Kit (74134) according to the manufacturer's instructions, quantified by NanoDrop and checked for integrity by capillary electrophoresis. One μg of the purified RNA was added of ERCC RNA Spike-In Mix 2 (Thermo Fisher Scientific #4456740, 0.5–1 μl of a 1:100 dilution per sample) and the total volume brought to 50 μl with UltraPure Water (Thermo Fisher Scientific #10977-023). mRNA purification, RNA fragmentation, first and second strand cDNA synthesis were performed according to the TruSeq RNA Sample Preparation v2 Kit (Illumina RS-122-2001) using Superscript III for reverse transcription instead of Superscript II (incubation time: 50°C for 50 min). cDNA was purified with AMPure XP beads and eluted in 27.5 μl 10 mM Tris-HCl pH 8.0, 25 μl of which were transferred to

a new tube and subjected to a NEBNext Ultra II DNA Library Prep Kit for Illumina (NEB #E7645), using half of the recommended reagents' volumes. The NEBNext Adaptor for Illumina was diluted 1:5 in Tris/NaCl, pH 8.0 (10 mM Tris-HCl pH 8.0, 10 mM NaCl) and the ligation step extended to 30'. Library enrichment was performed with the KAPA HotStart PCR kit (Roche Diagnostics KK2502) in 50 µl of total reaction volume (10 µl 5× KAPA buffer, 1.5 µl 10 mM dNTPs, 0.5 µl 10 µM NEB Universal PCR primer, 0.5 µl 10 µM NEB index primer, 1 µl KAPA polymerase, 16.5 µl nuclease-free water, and 20 µl sample). Samples were enriched with 8 PCR cycles (98°C, 45"; [98 C, 15"; 60°C, 10"] × 9; 72°C, 1'; 4°C, hold), purified with 0.9 volumes of AMPure XP PCR purification beads and eluted with 33 µl of 10 mM Tris-HCl pH 8.0. Library concentration, quality, and fragment size were assessed by Qubit fluorometric quantification (Qubit dsDNA HS Assay Kit, InvitrogenTM Q32851) qPCR and Fragment analyzer. Multiplexed libraries were pooled and sequenced on the Illumina NovaSeq 6000 platform (50–100 bp, paired end reads) at the Vincent J. Coates Genomics Sequencing Laboratory at UC Berkeley, supported by NIH S10 OD018174 Instrumentation Grant.

In the second batch (3 replicates each of WT 786-O cells, HIF-1α O.E., HIF-2α/1α, HIF-2α O.E., HIF-1α/2α) total RNA was extracted with TRIzol (Thermo Fisher Scientific #15596026) according to the manufacturer's instructions, performing an additional wash with 1 volume of chloroform after the recommended phenol:chloroform extraction. RNA was quantified by NanoDrop and checked for integrity by capillary electrophoresis. Five µg of total RNA were DNase treated (Ambio #AM1906) and 1 µg of DNase-free RNA was subject to poly-A purification and library preparation with the NEBNext Poly(A) mRNA Magnetic Isolation Module (NEB #E7490S) in combination with the NEBNext Ultra II RNA Library Prep Kit for Illumina (NEB #E7770S). The NEBNext Adaptor for Illumina was diluted 1:5 in Tris/NaCl, pH 8.0 (10 mM Tris-HCl pH 8.0, 10 mM NaCl) and the ligation step extended to 30'. Libraries were enriched with 8 PCR cycles. Library concentration was assessed by Qubit quantification (Qubit dsDNA HS Assay Kit, InvitrogenTM Q32851). Multiplexed libraries were pooled and sequenced on the Illumina NovaSeq 6000 platform (150 bp, paired end reads) by MedGenome Inc (Foster City, CA).

## RNA-seq analysis

RNA-seq raw reads were quality-checked with FastQC (http://www.bioinformatics.babraham.ac.uk/projects/fastqc) and aligned onto the human genome (hg38) using STAR 2.6.1d RNA-seq aligner (*Dobin et al., 2012*) with the following options: `--outSJfilterReads Unique --outFilterMultimapNmax 1 --outFilterIntronMotifs RemoveNoncanonical --outSAMstrandField intronMotif`. We used Samtools 1.9 (*Li et al., 2009*) to convert STAR output.sam files into .bam files, and to sort and index them. We then counted how many reads overlapped an annotated gene (GENECODE v32 annotations) using HTSeq 0.11.0 (*Anders et al., 2014*) (`htseq-count --stranded=no -f bam --additional-attr=gene_name -m union`), and used the output counts files to find differentially expressed genes with DESeq2 (*Love et al., 2014*), run with default parameters within the Galaxy platform (*Blankenberg et al., 2010*; *Giardine et al., 2005*; *Goecks et al., 2010*). Differentially expressed genes (DEGs) were called using an adjusted p-value ≤0.01, a fold change ≥2 and ≥0.5 mean counts. Gene transcript levels were visualized on the hg38 genome with the IGV (*Robinson et al., 2011*; *Thorvaldsdóttir et al., 2013*) using the bigWig output files from deepTools bamCoverage (`--binSize 50 --extendReads 250 –normalizeUsingRPKM`).

## Datasets and accession numbers

All high-throughput sequencing data generated in this publication have been deposited in NCBI's Gene Expression Omnibus (*Edgar et al., 2002*) and are accessible through GEO Series accession number GSE207575. SPT raw data are accessible through DOI: 10.5281/zenodo.7317360.

## Acknowledgements

We thank Luke Lavis for providing fluorescent HaloTag ligands; the CRL Flow Cytometry Facility for assistance with cell sorting; the QB3 High Throughput Screening Facility for providing access to the Opera Phenix automated confocal microscope; Dr. Alec Heckert for sharing data analysis pipeline and usage instructions; Dr. Hatice Kaya-Okur for valuable suggestions on Cut&Run protocols. Dr. Alec Heckert, Dr. Andrew Belmont, Dr. John Lis, Dr. Jiang Xu, Dr. Max Staller, Dr. Shasha Chong, and members of Tjian/Darzacq labs for helpful discussions and critical reading of the manuscript. This work was supported by the NIH grant U54-CA231641-01 (to XD), and the Howard Hughes Medical Institute

(to RT). This work used the Vincent J. Coates Genomics Sequencing Laboratory at UC Berkeley, supported by the NIH S10 OD018174 Instrumentation Grant.

## Additional information

### Competing interests
Robert Tjian: is one of the three founding funders of eLife, a member of eLife's Board of Directors and a co-founder of Eikon Therapeutics. Xavier Darzacq: is a co-founder of Eikon Therapeutics. The other authors declare that no competing interests exist.

### Funding

| Funder | Grant reference number | Author |
| --- | --- | --- |
| National Institutes of Health | U54-CA231641-01 | Xavier Darzacq |
| Howard Hughes Medical Institute | | Robert Tjian |

The funders had no role in study design, data collection and interpretation, or the decision to submit the work for publication.

### Author contributions
Yu Chen, Conceptualization, Software, Formal analysis, Validation, Investigation, Visualization, Methodology, Writing - original draft, Writing – review and editing; Claudia Cattoglio, Formal analysis, Investigation, Visualization, Writing – review and editing; Gina M Dailey, Qiulin Zhu, Resources, Investigation; Robert Tjian, Conceptualization, Funding acquisition, Writing – review and editing; Xavier Darzacq, Conceptualization, Supervision, Funding acquisition, Writing – review and editing

### Author ORCIDs
Yu Chen http://orcid.org/0000-0001-7856-4648
Claudia Cattoglio http://orcid.org/0000-0001-6100-0491
Gina M Dailey http://orcid.org/0000-0002-8988-963X
Robert Tjian http://orcid.org/0000-0003-0539-8217
Xavier Darzacq http://orcid.org/0000-0003-2537-8395

### Decision letter and Author response
Decision letter https://doi.org/10.7554/eLife.75064.sa1
Author response https://doi.org/10.7554/eLife.75064.sa2

## Additional files

### Supplementary files
• Supplementary file 1. Summary of sample sizes and estimated key parameter values for all fSPT experiments.
• Supplementary file 2. Coordinates of HIF-1α responsive sites.
• Supplementary file 3. Coordinates of HIF-2α responsive sites.
• Transparent reporting form

### Data availability
NGS data have been deposited in NCBI's GEO under accession number GSE207575. SPT raw data are accessible through https://doi.org/10.5281/zenodo.7317360.

The following datasets were generated:

| Author(s) | Year | Dataset title | Dataset URL | Database and Identifier |
|---|---|---|---|---|
| Cattoglio C, Chen Y, Dailey G, Zhu Q, Tjian R, Darzacq X | 2022 | Mechanisms Governing Target Search and Binding Dynamics of Hypoxia-Inducible Factors | https://www.ncbi.nlm.nih.gov/geo/query/acc.cgi?acc=GSE207575 | NCBI Gene Expression Omnibus, GSE207575 |
| Chen Yu | 2021 | Mechanisms Governing Target Search and Binding Dynamics of Hypoxia-Inducible Factors | https://zenodo.org/record/7317360 | Zenodo, 10.5281/zenodo.7317360 |

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

## Appendix 1

### Analysis of source of variations

Since we observed huge cell-to-cell variation (*Figure 2A–B*), we used bootstrapping to determine the source of variation, and decide how many cells, replicates, and trajectories will be needed to get a relatively good estimate for the parameters.

We used Halo-HIF-2α KIN clone A31 for the bootstrapping, as this is the cell line where the most data were collected. We used 243 movies (each movie corresponds to one cell) collected from 11 different replicates (each replicate was done on a different day) as the pool to draw sample from. We performed sampling in three different ways. First, we pooled all trajectories from all cells and randomly drawn different number of trajectories (i.e. sample size n=10, 25, 50, 100, 200, 400, 800, 1600, 3200, and 6400) from the pool with replacement. After each draw, the resulting trajectories were used for SA analysis to generate the diffusion spectrum and then the fraction bound was calculated. We performed 100 draws for each of the sampling size and plotted box plot on the resulting bound fraction estimation to see variation. This way, we can estimate if we randomly sample n trajectories 100 times, how variable the result will be. The result was as expected – the more trajectories we draw, the less variable the estimation is between the 100 times of drawing when looking at the bound fraction (*Figure 2—figure supplement 3A*, left).

Second, instead of sampling a mixture of trajectories from all 243 cells, we randomly picked N number of unique cells (N=1, 10, 20, 40, 60, 80, 100, 120) from the 243 cells, pooled the trajectories from these selected cells, and sampled from these pooled trajectories. The sample size and number of draws were kept the same. This way, we can compare if we randomly take only N cells from the population (243 cells) to estimate the bound fraction, do that 100 times for each sample size, how bad the variation is compared to if we take all 243 cells. This will be a good estimation of how many cells we need to pick. Certainly, small number of cells gives smaller number of total trajectories and if we draw large number of trajectories from them (such as n=6400) it will be over-sampled. Our result shows that when we pick as many as 60 cells, the difference of estimated bound fraction from these 60 cells compared to when estimated from all 243 cells can be less than 5% when sampling more than 800 trajectories (*Figure 2—figure supplement 3A*, middle; and *Figure 2—figure supplement 3B*).

Third, since different replicates were performed on different days, we use day as the unit to group trajectories. We randomly picked 1 day (i.e. 1 replicate), combined all trajectories from Halo-HIF-2α KIN clone A31 from that day, which normally is ~20 cells, and use this mixture of trajectories for sampling. This way, we can compare how variable our estimation will be from replicate to replicate. Indeed 1 replicate is not good enough to give good estimation as the variation is still high even if we sample large number of trajectories (*Figure 2—figure supplement 3A*, right). However, if we randomly pick 3 days instead of 1, we reduce the variance, and the estimated bound fraction from these 3 days compared to when estimated from all 243 cells from 11 days can be less than 5% when sampling more than 800 trajectories (*Figure 2—figure supplement 3C*).

From our bootstrapping results, we think an estimation of bound fraction from 3 replicates with 20 cells each replicate (and usually we can collect total more than 2500 trajectories from 3 replicates) can be good enough, at least for comparing different conditions such as WT versus mutants, where a difference of 10% bound fraction was usually observed.

## Appendix 2

### Using fluorescence intensity as approximation for protein expression level

Since we have shown that stoichiometry may affect protein behavior, to compare the molecular dynamics across different HIF-α forms that were exogenously expressed, it is important to control for similar expression levels. Unfortunately, we observed variegated expression levels with the same L30 promoter even with stable cell lines. As a result, for exogenously expressed Halo-proteins in *Figure 4* and *Figure 5*, we used fluorescence intensity as approximation for protein expression levels to guide our selection of which cell to image.

During imaging, cells were doubly labeled – one channel with excess JFX549 dye, which was used for localizing cells and define ROI, the other channel with limited JFX646 dye for fSPT (*Figure 1D*). The expression level for the Halo-tagged protein can be estimated by looking at fluorescence intensity from the JFX549 channel. Since we do not perform any bleaching before collecting SPT movies, the initial localization density (i.e. average number of molecules detected per ROI in the first 10 frames of the tracking experiment) from the JFX646 channel can also reflect the concentration of Halo-tagged proteins. Because first, for quantification of initial localization density, we only averaged the first 10 frames of each movie (corresponding to the first 50 ms) so that impact from photobleaching is neglectable, and second, with our fast-acquisition rate, we are capturing all molecules in the focal plane, including the fastest-moving population.

For Halo-HIF-1β and Halo-HIF-2α KIN lines, because the endogenous HIF-1β expresses at a much higher concentration than HIF-2α, when cells were fully labeled with JFX549, the fluorescence level for Halo-HIF-1β KIN cells were much higher than Halo-HIF-2α KIN cells. Similarly, when stained with same concentration of JFX646 for these two cell lines, the overall Halo-HIF-1β localization density was much higher compared to Halo-HIF-2α, consistent with much higher expression levels (*Figure 4—figure supplement 1A*). However, for SPT, a low localization density is needed to minimize tracking error. To keep localization density low, for *Figure 2*, Halo-HIF-1β had to be stained at ~1/10 of the JFX646 concentration used for Halo-HIF-2α (*Figure 4—figure supplement 1B*, 0.3125 or 0.625 nM instead of 5 nM) to achieve ideal localization density that is similar to Halo-HIF-2α (orange dots). This shows that for different expression levels, an order-of-magnitude difference in dye concentration needs to be used to get similar localization density. In addition, we found that while 5 nM gives very high localization density, when stained at a low concentration within a factor of 2 (0.3125 versus 0.625 nM), the resulting localization density for Halo-HIF-1β was also roughly the same. This shows that with similar dye concentration (within a factor of 2), localization density can be a rough approximation for protein levels.

We then utilize the localization density as a quality control for experiments where expression level needs to be kept similarly. For the L30-driven Halo-HIF-α expression, the variation of expression level is big from cell to cell and we are able to find cells with similar expression levels (close to endogenous) across different HIF-α variants. Thus, we used the JFX549 channel to pick cells with similar fluorescence intensity (usually the lowest intensity, which is the closest to endogenous expression level) for imaging, after cells were fully labeled with JFX549. Because these cells were also stained with similar level of JFX646 dye for fSPT when a set of experiments with multiple conditions were compared, cells picked for similar JFX549 intensity usually have similar localization densities in the JFX646 channel. As confirmed by quantification in *Figure 4—figure supplement 1D–G*, we did have within twofold differences in localization density for different conditions in each set of experiments performed, indicating that for each comparison, the expression level across different conditions were kept roughly similar.

