## [Editor Report]

This work on dissecting the function of transcription factors using single-molecule methods is likely to appeal to the broad *eLife* readership and the gene regulation community at large. In particular, the role of transcription activation domains in transcriptional specificity is a timely contribution to the field.

---

## [Decision Letter]

**Decision letter after peer review:**

Thank you for submitting your article "Mechanisms Governing Target Search and Binding Dynamics of Hypoxia-Inducible Factors" for consideration by *eLife*. Your article has been reviewed by 3 peer reviewers, one of whom is a member of our Board of Reviewing Editors, and the evaluation has been overseen by Kevin Struhl as the Senior Editor. The following individual involved in review of your submission has agreed to reveal their identity: David M Suter (Reviewer #3).

Essential revisions:

1. As you can see below, there was broad enthusiasm for the single-molecule imaging approach for uncoupling DNA binding domains and trans-activation domains. However, there was also a consensus that the key potential advance of this paper is connecting reaction diffusion kinetics to the specificity of DNA binding. Yet, the specificity aspect, although given as a primary motivation, is not adequately established experimentally. Such data are essential to include in a revision.

2) Analysis of single-molecule tracking data is central to the claims, and there were a number of concerns raised by each referee about the choice of algorithms, models, use of controls, types of mobility, etc. These issues need to be addressed in detail in a revised manuscript.

*Reviewer #1 (Recommendations for the authors):*

1. The authors motivate the work with the fact that different HIF1 isoforms have similar in vitro binding specificity but different biological activity. Yet, they don't ultimately resolve this question or even shed much light on it. For example, nothing in the present analysis says whether the different dimer moieties actually bind different sequences (for example ChIP-seq or in vitro gel shift). As such, the claim of specificity being determined by IDRs is a little oblique. Dynamics are clearly determined by IDRs as the authors nicely show, but the connection back to binding specificity is very indirect. However, this group could easily carry out gel shift binding assays or even single-molecule kinetic assays in vitro to solidify this point.

2. All the conclusions are essentially reached exclusively through the SPT assay with a new analysis approach that has not yet been peer-reviewed. As such, there is not much to latch on to from a referee perspective. Either an orthogonal assay or comparison to existing approaches of SPT analysis would greatly increase the confidence in the results.

*Reviewer #2 (Recommendations for the authors):*

The authors should improve their manuscript by including the following points:

– They should quantify the expression levels of all endogenous fusion proteins and exogeneously expressed transcription factors and mutants by several independent Western Blots, otherwise it is hard to judge whether a change in bound fraction or peak diffusion coefficient is due to the condition tested or rather due to differences in expression levels.

– They should control for proper gene transcription of their mutants, in particular the domain-swap mutants, analogous to the experiments using the luciferase reporter gene for HIF-2a.

– They should control that their high laser power of 1100 mW for illumination does not harm the cells.

– They should perform more measurements to arrive at statistically sound analysis. in particular, more biological replicates should enter each determination of bound fraction or peak diffusion coefficient.

– In addition to bound fractions, the authors should quantify the peak diffusion coefficient for each mutant and experimental condition and compare the relative changes between bound fraction and diffusion coefficient for each mutant and experimental condition, to allow estimating which property is more important.

The authors could further improve the impact of their manuscript by including the following points:

– For now, the authors only characterise the effect of different domains of HIF-a on HIF-1b or HIF-a binding. It would be informative if the authors also could characterise the effect of HIF-1b domains on its bound fraction and diffusion properties.

– Previously, the confinement of single particles was analysed (Lerner et al., Mol Cell 2020). The extent of confinement was shown to depend on the intrinsicall disordered domain of glucocorticoid receptors (Garcia et al., Mol Cell 2021). It would be good to see to which extent the intrinsically disordered domain of HIF mediates different confinement states, and whether there are differences between the IDRs of HIF-1a and HIF-2a.

– To answer the question whether IDRs actually contribute to differential DNA sequence specificity, the authors could perform ChIP experiments analogous to Figure S2D for the domain-swapped mutants of HIF-a.

Additional points:

– Line 145-158: analysis of diffusion behaviour using non-parametric Bayesian approach: the authors use a novel method to analyse their data. Although they provide a reference, there need to be more details in the text on how this method works

– Figure S4B: it is unclear which cell line was used, an unedited cell line with endogeous HIF-2a or a Halo-HIF-2a edited cell line?

– Line 247: …we do not observe a significant change in the overall speed of the diffusing population.

– Diffusion does not have a speed, please use other wording.

– Figure 1D, E: missing scale bar description.

– What is does V5 refer to?

– Line 92: please correct: HIF isoforms.

– Line 923: should read "as in Figure 4".

*Reviewer #3 (Recommendations for the authors):*

Overall this is a solid piece of work that convincingly shows the role of transactivation domains in determining the TF bound fraction. Although the impact of TADs on search is not entirely new, the dissection performed here using state of the art SPT and analysis is very valuable.

I do have a few suggestions that may further strengthen and enrich their findings, which mainly imply further analysis of their already acquired datasets:

1. The authors show that the different TADs of the α subunits have a differential impact on the bound fraction of HIF-1beta. They suggest that this is due to a difference in search capacity. Are the authors implying differences in on-rates? The bound fraction will be influenced both by on-rates and off-rates.

I suggest that using their already acquired data, the authors compute and show the off-rates (or dwell times) for all the TFs they analyze. In addition, I suggest they also compute a pseudo on-rate (as in Raccaud et al. 2019 for example), which they can easily do from their data as the relative concentration of each TF they study is already captured from their data. This would provide more information on the relative contribution of on- vs off-rate to the bound fraction.

2. Concerning the stoichiometry differences between the β and α subunits, I do agree that this is consistent with the western blots but these are not very quantitative (and also the band intensities were not quantified). The authors could perform image-based quantification using epifluorescence microscopy with a Halo dye for a more precise quantification.

3. It would be valuable to classify the diffusive behaviours of the different TF they looked at in brownian, super-diffusion and sub-diffusion behaviour as these may reflect the size of the complex they are part of.

---

## [Author Response]

Essential revisions:1. As you can see below, there was broad enthusiasm for the single-molecule imaging approach for uncoupling DNA binding domains and trans-activation domains. However, there was also a consensus that the key potential advance of this paper is connecting reaction diffusion kinetics to the specificity of DNA binding. Yet, the specificity aspect, although given as a primary motivation, is not adequately established experimentally. Such data are essential to include in a revision.

We thank the reviewer for pointing this out. In the revised manuscript, we have included new Cut&Run and RNA-seq data to confirm IDR-specific binding and gene activation using orthogonal assays from SPT.

2) Analysis of single-molecule tracking data is central to the claims, and there were a number of concerns raised by each referee about the choice of algorithms, models, use of controls, types of mobility, etc. These issues need to be addressed in detail in a revised manuscript.

We have included our response to the specific concerns below.

Reviewer #1 (Recommendations for the authors):1. The authors motivate the work with the fact that different HIF1 isoforms have similar in vitro binding specificity but different biological activity. Yet, they don't ultimately resolve this question or even shed much light on it. For example, nothing in the present analysis says whether the different dimer moieties actually bind different sequences (for example ChIP-seq or in vitro gel shift). As such, the claim of specificity being determined by IDRs is a little oblique. Dynamics are clearly determined by IDRs as the authors nicely show, but the connection back to binding specificity is very indirect. However, this group could easily carry out gel shift binding assays or even single-molecule kinetic assays in vitro to solidify this point.

We thank the reviewer for raising this important point. When we talk about sequence specificity, we refer to DNA sequence specificity, traditionally validated by *in-vitro* binding experiments. However, inside a living cell, TFs do not bind naked DNA, but rather land on chromatin fibers composed of DNA and various nuclear proteins. It is already known that outside their native chromatin environment HIF-1α and 2α bind to the same DNA sequence (Schödel et al., 2011; Wenger et al., 2005). Even luciferase reporter assays are misleading, as HIF-1α can also activate reporter expression via an EPO enhancer, which only responds to HIF-2α in vivo (Varma and Cohen, 1997; Warnecke et al., 2004; Rankin et al., 2007). Because of the concerns raised by reviewers, we decided to address the specificity question with genomic approaches complementary to SPT measurements in vivo. We thus performed Cut&Run on cells expressing either HIF-1α or -2α isoforms and in parallel analyzed their RNA expression profile. Our new data show that, although by Cut&Run analysis most of the HIF binding sites are shared between 1α and 2α, there are numerous specific subsets of binding sites (and genes) that clearly respond differentially to HIF-1α vs -2α levels. We have also included Cut&Run and RNA-seq results of HIF-α domain-swap variants (HIF-1α/2α or HIF-2α/1α), showing that the IDR indeed contributes to the target specificity of HIF-1α and -2α (Figure 6).

2. All the conclusions are essentially reached exclusively through the SPT assay with a new analysis approach that has not yet been peer-reviewed. As such, there is not much to latch on to from a referee perspective. Either an orthogonal assay or comparison to existing approaches of SPT analysis would greatly increase the confidence in the results.

We note that the new SPT analysis pipeline has now been peer-reviewed and accepted for publication at *eLife*. Additionally, we now include genomic assays as an orthogonal approach to SPT that further strengthens our conclusions.

Reviewer #2 (Recommendations for the authors):The authors should improve their manuscript by including the following points:– they should quantify the expression levels of all endogenous fusion proteins and exogeneously expressed transcription factors and mutants by several independent Western Blots, otherwise it is hard to judge whether a change in bound fraction or peak diffusion coefficient is due to the condition tested or rather due to differences in expression levels.

See response to Reviewer #2’s Public Review. We used localization density to roughly control for protein expression at a single cell level, while an ensemble assay like Western Blot is of limited use when dealing with polyclonal cell lines.

– They should control for proper gene transcription of their mutants, in particular the domain-swap mutants, analogous to the experiments using the luciferase reporter gene for HIF-2a.

See response to Reviewer #2’s Public Review. We have included RNA-seq data in cells expressing domain swaps and WT counterparts, showing all these variants are transcriptionally active.

– They should control that their high laser power of 1100 mW for illumination does not harm the cells.

See response to Reviewer #2’s Public Review. Briefly, our imaging scheme is unlikely to generate phototoxicity artifacts. Also, we are comparing results across all conditions with the exact same imaging set-up, so potential artifacts should be accounted and controlled for.

– They should perform more measurements to arrive at statistically sound analysis. in particular, more biological replicates should enter each determination of bound fraction or peak diffusion coefficient.

See response to Reviewer #2’s Public Review. We combined data from 3 replicates to estimate the parameters reported in figures. Since the replicate-to-replicate variation is much less than the effect of mutants or drug perturbation, we think it does not affect our main conclusions.

– In addition to bound fractions, the authors should quantify the peak diffusion coefficient for each mutant and experimental condition and compare the relative changes between bound fraction and diffusion coefficient for each mutant and experimental condition, to allow estimating which property is more important.

See response to Reviewer #2’s Public Review. We have included these data in Supplement File 1. However, it is hard to define “importance” when comparing bound fraction and diffusion coefficient. Specifically, it is hard to tell how much change in the bound fraction is equally “important” to how much change is observed in the diffusion coefficient. All we can measure is the effect on transcription regulation of the combination of the two.

The authors could further improve the impact of their manuscript by including the following points:– for now, the authors only characterise the effect of different domains of HIF-a on HIF-1b or HIF-a binding. It would be informative if the authors also could characterise the effect of HIF-1b domains on its bound fraction and diffusion properties.

Since the main interest of this paper is to dissect isoform-specific regulation and given that HIF-1β is a shared partner, we did not focus on HIF-1β domains. This can be an interesting follow-up study.

– Previously, the confinement of single particles was analysed (Lerner et al., Mol Cell 2020). The extent of confinement was shown to depend on the intrinsicall disordered domain of glucocorticoid receptors (Garcia et al., Mol Cell 2021). It would be good to see to which extent the intrinsically disordered domain of HIF mediates different confinement states, and whether there are differences between the IDRs of HIF-1a and HIF-2a.

For these particular cell lines we had to use a non-photoactivatable dye for imaging and we kept the localization density very low for each cell in order to avoid trajectories misconnections during tracking. As a result, we do not have a lot of trajectories per cell, and most of the trajectories are very short (average length ~ 3 frames) so we do not have enough data from each cell to perform the confinement analysis.

We agree that this is a potentially interesting question, but we have chosen to include the most essential and directly relevant data that is needed for our current conclusions which already includes a vast amount of data. We believe this and other questions are best left for subsequent follow-up studies.

– To answer the question whether IDRs actually contribute to differential DNA sequence specificity, the authors could perform ChIP experiments analogous to Figure S2D for the domain-swapped mutants of HIF-a.

See response to Reviewer #2’s Public Review. We have now included Cut&Run and RNA-seq data showing IDR-specific binding and gene activation.

Additional points:– Line 145-158: analysis of diffusion behaviour using non-parametric Bayesian approach: the authors use a novel method to analyse their data. Although they provide a reference, there need to be more details in the text on how this method works

We have updated the manuscript to include more details.

– Figure S4B: it is unclear which cell line was used, an unedited cell line with endogeous HIF-2a or a Halo-HIF-2a edited cell line?

These were WT 786-O cells. We observed similar results for Halo-HIF-2α edited cells.

– Line 247: …we do not observe a significant change in the overall speed of the diffusing population.– Diffusion does not have a speed, please use other wording.

Changed to “diffusion coefficient” / “diffusion rate”.

– Figure 1D, E: missing scale bar description.

Scale bar (2 μm) added.

– What is does V5 refer to?

V5 tag is a short peptide tag for detection / purification of proteins. In our revised manuscript, we used V5 tag as the epitope for Cut&Run.

– Line 92: please correct: HIF isoforms.

Corrected.

– Line 923: should read "as in Figure 4".

Corrected, thank you.

Reviewer #3 (Recommendations for the authors):Overall this is a solid piece of work that convincingly shows the role of transactivation domains in determining the TF bound fraction. Although the impact of TADs on search is not entirely new, the dissection performed here using state of the art SPT and analysis is very valuable.I do have a few suggestions that may further strengthen and enrich their findings, which mainly imply further analysis of their already acquired datasets:1. The authors show that the different TADs of the α subunits have a differential impact on the bound fraction of HIF-1beta. They suggest that this is due to a difference in search capacity. Are the authors implying differences in on-rates? The bound fraction will be influenced both by on-rates and off-rates.I suggest that using their already acquired data, the authors compute and show the off-rates (or dwell times) for all the TFs they analyze. In addition, I suggest they also compute a pseudo on-rate (as in Raccaud et al. 2019 for example), which they can easily do from their data as the relative concentration of each TF they study is already captured from their data. This would provide more information on the relative contribution of on- vs off-rate to the bound fraction.

We understand that the bound fraction is determined by both on- and off-rates. However, because 1) the current fast SPT imaging data does not have the capability to measure reliably the off-rates, and 2) we cannot assume a single off-rate for all HIF variants, or even a single HIF protein, we are not able to measure the on-rates for these proteins. We have changed our wording accordingly to be more precise and only imply that IDR impacts both HIF target search and binding, but we are hesitant to go further than that to imply the differences are mainly caused by modulating on-rates.

We did try slow SPT (500-ms and 1-s frame rate) to measure the residence time (or off-rates), but HIF-2α binds remarkably stably, with RTs that are almost indistinguishable from the H2B control (used as a photobleaching control). As a result, unfortunately, we are not able to reliably estimate the actual off-rate. We have also tried FRAP, but these L30-driven proteins are expressed at very low levels (near background fluorescence level on our Airyscan confocal microscope), which would require substantial optimization to be able to get reliable data.

As a result, due to technical challenges we would like to hold-off deciding the relative contribution of on- vs off-rates to the bound fraction to speed up publication. We have added this to the discussion to clarify what we are trying to claim and what we are not. We do hope to have a follow-up study addressing this question.

2. Concerning the stoichiometry differences between the β and α subunits, I do agree that this is consistent with the western blots but these are not very quantitative (and also the band intensities were not quantified). The authors could perform image-based quantification using epifluorescence microscopy with a Halo dye for a more precise quantification.

See Figure 4—figure supplement 1A, B. We estimated protein expression level for HIF-1β and 2α with localization density for cells imaged. Again, bulk analysis such as western blotting does not reflect the actual protein level of the cells selected for imaging. Our image-based single cell analysis results also show that 1β was expressed at a much higher level than 2α.

3. It would be valuable to classify the diffusive behaviours of the different TF they looked at in brownian, super-diffusion and sub-diffusion behaviour as these may reflect the size of the complex they are part of.

Yes, this is indeed a very interesting question to ask but we believe this and other questions are beyond the scope of this study and better left for follow-up studies.